



# Atmospheric O$_2$ and CO$_2$ measurements at a single height provide weak constraint on surface carbon exchange

Kim A. P. Faassen[1], Jordi Vilà-Guerau de Arellano[1,2], Raquel González-Armas[1], Bert G. Heusinkveld[1], Ivan Mammarella[3], Wouter Peters[1,4], and Ingrid T. Luijkx[1]

[1]Meteorology and Air Quality, Wageningen University and Research, Wageningen, the Netherlands
[2]Atmospheric Chemistry Department, Max Planck Institute for Chemistry, 55128 Mainz, Germany
[3]Institute for Atmospheric and Earth System Research (INAR) / Physics, Faculty of Science, University of Helsinki, Helsinki, Finland
[4]University of Groningen, Centre for Isotope Research, Energy and Sustainability Research Institute Groningen, Groningen, the Netherlands

**Correspondence:** Kim Faassen (kim.faassen@wur.nl)

**Abstract.** The ratios of atmospheric tracers are often used to interpret the local CO$_2$ budget, where measurements at a single height are assumed to represent local flux signatures. Alternatively, these signatures can be derived from direct flux measurements or using fluxes derived from measurements at multiple heights. In this study, we contrast interpretation of surface CO$_2$ exchange from tracer ratio measurements at a single height versus measurements at multiple heights.

Specifically, we analyse the ratio between atmospheric O$_2$ and CO$_2$ (exchange ratio, ER) above a forest canopy. We consider two alternative approaches: the exchange ratio of the forest (ER$_{forest}$) obtained from the ratio of the surface fluxes of O$_2$ and CO$_2$, derived from their vertical gradients measured at multiple heights, and the exchange ratio of the atmosphere (ER$_{atmos}$) obtained from changes in the O$_2$ and CO$_2$ mole fractions over time measured at a single measurement height. We investigate the diurnal cycle of both ER signals, with the goal to relate the ER$_{atmos}$ signal to the ER$_{forest}$ signal and to understand the

biophysical meaning of the ER$_{atmos}$ signal. We combined CO$_2$ and O$_2$ measurements from Hyytiälä, Finland during spring and summer of 2018 and 2019 with a conceptual land-atmosphere model and a theoretical relationship between ER$_{atmos}$ and ER$_{forest}$ to investigate the behavior of ER$_{atmos}$ and ER$_{forest}$ during different environmental conditions. We show that the ER$_{atmos}$ signal rarely directly represents the forest exchange, mainly because it is influenced by entrainment of air from the free troposphere into the atmospheric boundary layer. The influence of these larger scale signals leads to very high ER$_{atmos}$ values (even larger

than 2), especially in the early morning transition. These high values do not directly represent carbon cycle processes, but are rather a mixture of different signals. We show that the resulting ER$_{atmos}$ signal is not the average of the contributing processes, but rather an indication of the influence of large scale processes such as entrainment or advection. Our findings show that these processes are furthermore influenced by climate conditions, such as the 2018 heatwave, through their dependence on soil moisture and temperature.

We conclude that the ER$_{atmos}$ signal obtained from single height measurements rarely directly represents ER$_{forest}$ and therefore only provides a weak constraint on local scale surface CO$_2$ exchange, because large scale processes confound the signal. Single height measurements therefore always require careful selection of the time of day and should be combined with atmospheric





modelling to yield a meaningful representation of forest carbon exchange. More generally, we recommend to always measure at multiple heights when using multi-tracer measurements to study surface $CO_2$ exchange.

## 1 Introduction

Rising atmospheric carbon dioxide ($CO_2$) levels, resulting from fossil fuel combustion and land use change emissions and uptake by the terrestrial biosphere and oceans require a comprehensive assessment of the carbon exchange at local and global scales (Friedlingstein et al., 2022). Atmospheric oxygen ($O_2$) serves as a valuable tracer in enhancing our understanding of carbon exchange, due to the close linkage between $O_2$ and carbon dioxide ($CO_2$) in carbon cycle processes such as fossil fuel combustion, photosynthesis and respiration (Manning and Keeling, 2006; Worrall et al., 2013; Keeling and Manning, 2014; Bloom, 2015; Hilman et al., 2022). The Exchange Ratio (ER = -$O_2$/$CO_2$), denoted as the number of moles of $O_2$ exchanged per mole of $CO_2$ represents the specific link between $O_2$ and $CO_2$ for different processes (Keeling et al., 1998). Long-term $O_2$ and $CO_2$ measurements allow to derive the global ocean carbon sink (Stephens et al., 1998; Rödenbeck et al., 2008; Tohjima et al., 2019) and to estimate changes in fossil fuel emissions (Pickers et al., 2022; Ishidoya et al., 2020; Rödenbeck et al., 2023).

For global applications, a constant ER of 1.1 [mol mol$^{-1}$] is assumed for the terrestrial biosphere (Severinghaus, 1995). However, the ER of terrestrial biosphere exchange is not uniform at smaller scales; it varies between ecosystems and over time (Angert et al., 2015; Bloom, 2015; Battle et al., 2019; Hilman et al., 2022). Measuring the ERs of ecosystems and the underlying gross processes facilitates the partitioning of Net Ecosystem Exchange (NEE) into Gross Primary Production (GPP) and Total Ecosystem Respiration (TER) (Ishidoya et al., 2015; Faassen et al., 2023) which is still challenging (Reichstein et al., 2005). The ER for net ecosystem exchange can be determined from the ratio of the net turbulent surface fluxes of $O_2$ and $CO_2$ above the canopy, referred to as ER$_{forest}$ (see Figure 1). The $O_2$ surface fluxes can be inferred from the vertical gradient: the difference between $O_2$ mole fraction measurements at multiple heights, together with a turbulent exchange coefficient. Currently, available instruments so far have not allowed Eddy Covariance (EC) $O_2$ measurements. The ER$_{forest}$ signal predominantly represents forest exchange occurring in and below the canopy (small scale processes), comprising the individual ERs of TER (ER$_r$) and GPP (ER$_a$) (Ishidoya et al., 2013, 2015; Faassen et al., 2023). Alternatively, net ecosystem ERs have been estimated based on measurements of $O_2$ and $CO_2$ mole fractions in the atmosphere at a single height above the canopy. This is referred to as ER$_{atmos}$ (Figure 1) and is defined as the change in $O_2$ and $CO_2$ mole fractions over time (Seibt et al., 2004; Battle et al., 2019; Faassen et al., 2023).

In our recent study (Faassen et al., 2023), we showed a comprehensive comparison of the diurnal behaviour of ER$_{forest}$ and ER$_{atmos}$ using measurements collected above a boreal forest in Hyytiälä, Finland. Our analysis revealed that during the afternoon (the photosynthesis dominant period in Figure 1), the ER$_{atmos}$ signal approaches the ER$_{forest}$ value, although they did not converge completely. Furthermore, we showed that during the entrainment-dominant period (see Figure 1), the ER$_{atmos}$ signal strongly exceeded the expected ER value for biosphere exchange, which is typically around 1.1 (Severinghaus, 1995), and even



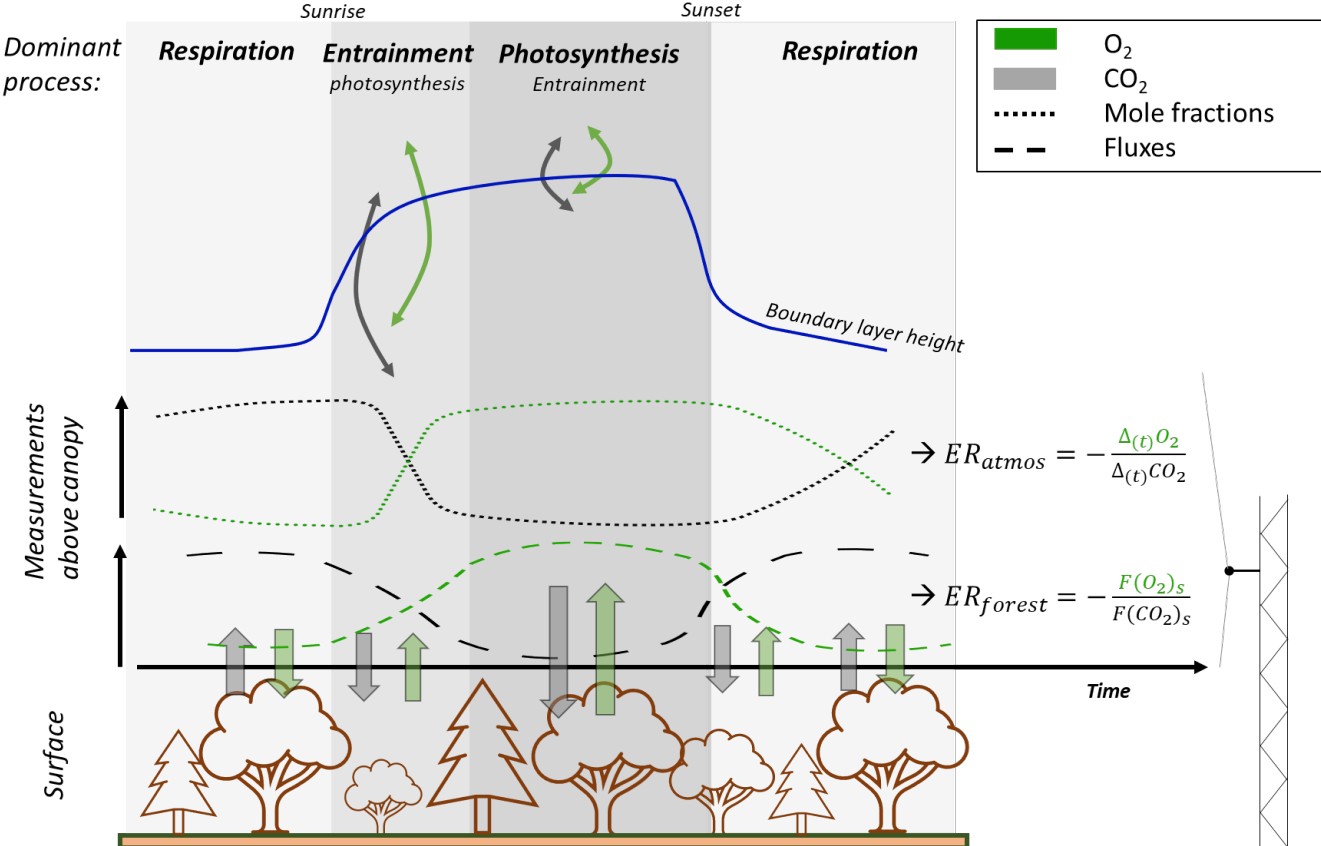

**Figure 1.** Schematic overview of the diurnal cycles of the surface fluxes and mole fractions of atmospheric $O_2$ and $CO_2$ above a forest canopy. The figure illustrates the dominant processes throughout the day, with forest exchange dominating the nocturnal and afternoon periods, while early morning signals are primarily influenced by entrainment of air from the residual layer or the free troposphere. The surface fluxes of $O_2$ and $CO_2$ result in the Exchange Ratio signal of the forest ($ER_{forest}$), while the changes in the mole fractions of $O_2$ and $CO_2$ over time lead to variations of the Exchange Ratio signal of the atmosphere ($ER_{atmos}$).

surpassed 2.0. Such high ER values ($>2.0$) cannot be attributed to a single process such as photosynthesis, respiration or fossil fuel combustion, as their ER values are below 2.0. We proposed that the high $ER_{atmos}$ signal was likely influenced by large scale processes, specifically the entrainment of air from the free troposphere into the boundary layer (Faassen et al., 2023). Also Seibt et al. (2004) and Yan et al. (2023) argue that $ER_{atmos}$ cannot capture the ER signal of a forest. In contrast, in the studies by Ishidoya et al. (2013, 2015) $ER_{forest}$ and $ER_{atmos}$ do result in similar values when small scale processes dominate over large scale processes. In Faassen et al. (2023) we concluded that an atmospheric model was needed to interpret the observed diurnal signals of $ER_{atmos}$ and $ER_{forest}$. The current study delivers this model-based analysis.





Until now atmospheric $O_2$ above forest canopies has primarily been modeled with relatively simple one-box models that
use only the surface components, lacking implementation of boundary layer dynamics such as entrainment and boundary layer
growth (Seibt et al., 2004; Ishidoya et al., 2013). Understanding how mole fractions, and consequently how $ER_{atmos}$ evolves
throughout the day requires accounting for these critical processes. Yan et al. (2023) recently modelled $O_2$ and $CO_2$ within and
below a canopy using a multi-layer model and showed that $ER_{atmos}$ and $ER_{forest}$ have diurnal and annual patterns. However,
$ER_{atmos}$ was treated as a constant value above the canopy and boundary layer dynamics were not accounted for. To expand
on the work by Yan et al. (2023) and gain further insight into the diurnal $ER_{atmos}$ behaviour above a canopy, in this study we
use the mixed layer model Chemistry Land-surface Atmosphere Soil Slab (CLASS) (Vilà-Guerau de Arellano et al., 2015).
In short, the model is able to represent the thermodynamics and biophysical processes associated with the diurnal variation
in the boundary layer and can provide insights into the processes contributing to $ER_{atmos}$ formation. Additionally, the model
facilitates the analysis of $ER_{atmos}$ behavior under more extreme conditions, that were not yet measured.


This study aims to enhance our understanding of single height $O_2$ and $CO_2$ measurements and the resulting $ER_{atmos}$ signal,
as observed above the canopy, and proposes a new relationship between the $ER_{atmos}$ and $ER_{forest}$ signal. We seek to determine
whether single height $O_2$ and $CO_2$ measurements could potentially be employed to estimate the ecosystem's ER despite the
mentioned limitations. Additionally, we explore whether the $ER_{atmos}$ signal allows to constrain boundary layer dynamics and
identify cases where large scale processes (e.g. entrainment of background air) influence the signal of small scale processes
(e.g. NEE) by analyzing different diurnal regimes of $ER_{forest}$ and $ER_{atmos}$. We combine measurements from campaigns in
Hyytiälä, Finland during the spring/summer of 2018 and 2019 with an analysis of the mixed layer model CLASS. This com-
bined approach allows us to address the following research questions: 1) When does $ER_{atmos}$ represent local forest exchange
processes, and become equal to $ER_{forest}$? 2) What is the underlying physical explanation for the high $ER_{atmos}$ values observed
in the recent study by Faassen et al. (2023)?

In this paper we first derive a theoretical relationship between $ER_{atmos}$ and $ER_{forest}$ that can help us to understand which
components influence the diurnal cycle of $ER_{atmos}$ and when $ER_{atmos}$ should indicate the same processes as $ER_{forest}$ (Sect. 2).
To evaluate the diurnal cycle of $ER_{atmos}$ we combine observational data with the model CLASS which are described in Sect. 3.
We show the model evaluation and the $ER_{atmos}$ and $ER_{forest}$ model results in Sect. 4, where we analyse different cases to explain
the diurnal behaviour of $ER_{atmos}$ during distinct periods of the day and investigate when $ER_{atmos}$ represents forest exchange. In
Sect. 5 we place our results in perspective and show how $ER_{atmos}$ should (not) be used. Finally, we present our conclusions on
the physical explanations for the differences between the diurnal behaviour of both $ER_{atmos}$ and $ER_{forest}$.





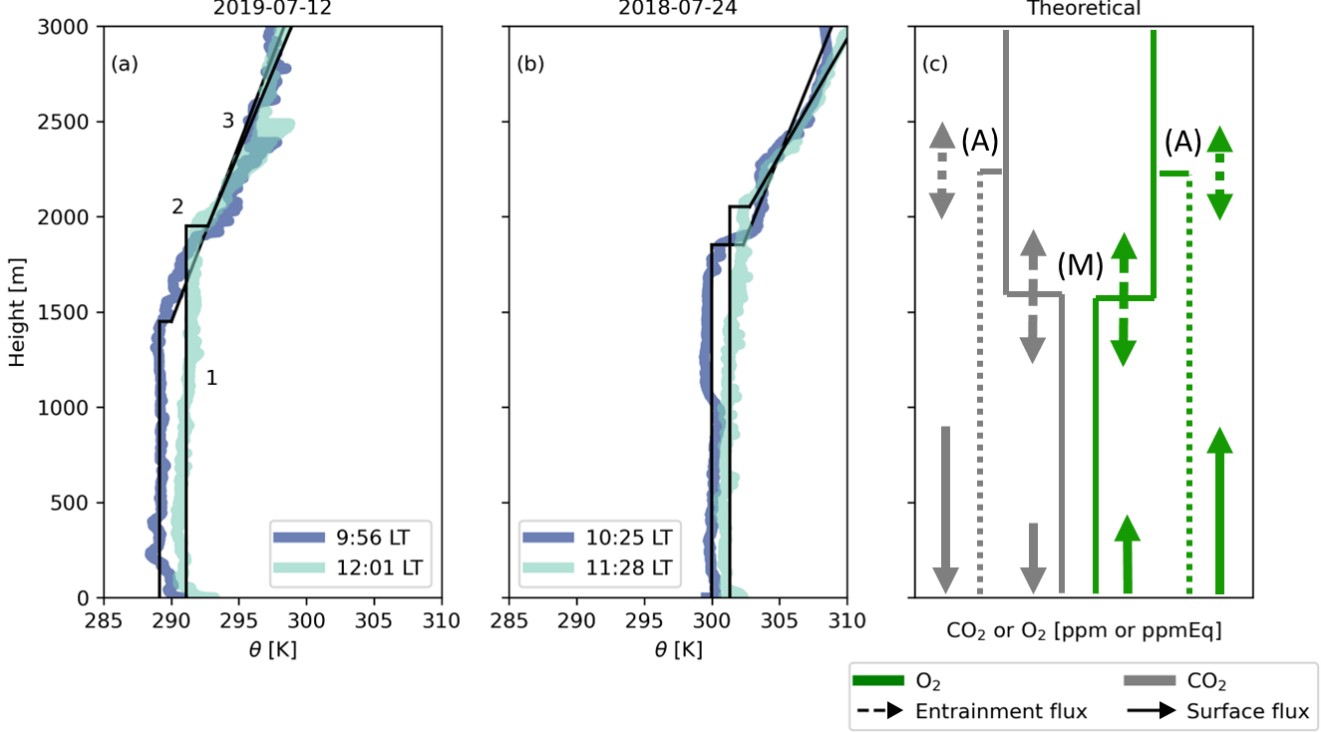

**Figure 2.** Vertical profiles of potential temperature ($\theta$) measured by radiosondes at Hyytiälä on 12 July 2019 (a) and 24 July 2018 (b). The observations are conceptualized (black lines) to show: 1) the well-mixed profiles at different time steps, 2) the jump between the boundary layer and the free troposphere, and 3) the lapse rate in the free troposphere. 1, 2 and 3 are used to initialize the CLASS model. (c) gives the theoretical vertical profiles of $O_2$ and $CO_2$ for the early morning (M) and late afternoon (A). The sizes of the arrows indicate the effects of entrainment (dashed lines) and the surface fluxes (solid lines) on the vertical profiles.

## 2 Fundamental concepts

### 2.1 The mixed layer theory

The land-atmosphere model CLASS (Vilà-Guerau de Arellano et al., 2015) is based on the mixed-layer theory which assumes that scalars (such as $O_2$, $CO_2$, $\theta$) are constant with height in the atmospheric boundary layer (Lilly, 1968; Tennekes, 1973). Figure 2 illustrates these assumptions for potential temperature ($\theta$), $O_2$ and $CO_2$. Within the mixed layer theory, no distinct surface layer exists, and a capping inversion links the mixed layer value (the bulk constant value) with the lapse rate of the free 100 troposphere. This inversion, termed the 'jump' ($\Delta_{(ft-bl)}$), represents the difference of a scalar (e.g. the $CO_2$ mole fraction) between the atmospheric boundary layer and the free troposphere. The free troposphere is represented by a linear change of the scalar with height (the lapse rate).




CLASS describes the well-mixed layer with a scalar constant in height (Figure 2). This scalar ($\phi$) can then be solved in the
mixed-layer with the following equations (Vilà-Guerau de Arellano et al., 2015):

$$\frac{\partial \phi}{\partial t} = \frac{(\overline{w'\phi'})_s - (\overline{w'\phi'})_e}{h} - adv(\phi) \tag{1}$$

where $\partial \phi / \partial t$ is the tendency (i.e. change over time) of a generic well-mixed scalar, $(\overline{w'\phi'})_s$ is the surface flux and represents
the small scale processes, $(\overline{w'\phi'})_e$ is the entrainment flux, $h$ is the boundary layer height and $adv(\phi)$ is the horizontal advection
of scalar $\phi$ into the well-mixed layer. $(\overline{w'\phi'})_e$ and $adv(\phi)$ represent the large scale processes, in contrast to the local surface
exchange $(\overline{w'\phi'})_s$.

The entrainment flux is dependent on the entrainment velocity and the jump:

$$(\overline{w'\phi'})_e = -w_e \cdot \Delta_{(ft-bl)}\phi = \left( \frac{\partial h}{\partial t} - w_{sub} \right) \cdot \Delta_{(ft-bl)}\phi \tag{2}$$

where $w_e$ is the entrainment velocity, $\Delta_{(ft-bl)}\phi$ is the jump between the free troposphere and the atmospheric boundary layer,
and $w_{sub}$ is the mean vertical subsidence velocity associated normally to high pressure systems, which we assume to be negli-
gible, because our focus does not lie on the influence of synoptic scale processes.

$\Delta_{(ft-bl)}\phi$ changes over time (see Figure 2) and depends on the surface fluxes and the air that is entrained from the free
troposphere (see Equation 1):

$$\frac{\partial \Delta_{(ft-bl)}\phi}{\partial t} = \gamma_\phi \cdot w_e - \frac{\partial \phi}{\partial t} \tag{3}$$

where $\gamma_\phi$ is the lapse rate of $\phi$ in the free troposphere.

Last, the growth of the boundary layer height ($h$) is of importance for the entrainment velocity and as a result the entrainment
flux of a certain scalar. The growth of the boundary layer is caused by the virtual potential temperature ($\theta_v$), also called
buoyancy:

$$\frac{\partial h}{\partial t} = -\frac{(\overline{w'\theta_v'})_e}{\Delta \theta_v} + w_s \tag{4}$$

where $\theta_v$ is the virtual potential temperature (i.e. potential temperature of dry air) and $w_s$ is the subsidence velocity. For more
details on these equations, see Vilà-Guerau de Arellano et al. (2015) and Sect. 3.2.2 and Sect. A2 for the application of $O_2$.

## 2.2   Theoretical relationship between ER$_{atmos}$ and ER$_{forest}$

The ER signal of the forest (ER$_{forest}$) is defined as (Faassen et al., 2023):

$$ER_{forest} = -\frac{(F_{O_2})_s}{(F_{CO_2})_s} \tag{5}$$



where $(F_{O_2})_s$ and $(F_{CO_2})_s$ are the mean net turbulent surface fluxes of $O_2$ and $CO_2$ respectively over a certain time period above the canopy, and can be derived from the vertical gradient of $O_2$ and $CO_2$ measurements at two heights (Faassen et al., 2023). Note that here we write the surface fluxes for both $O_2$ and $CO_2$ as $F_\phi$ instead of the general form $\overline{(w'\phi')}_s$ that was used above for the general theory.

The ER signal of the atmosphere ($ER_{atmos}$) is defined as (Faassen et al., 2023):

$$ER_{atmos} = -\frac{\partial O_2/\partial t}{\partial CO_2/\partial t} \approx -\frac{\Delta_{(t)}O_2}{\Delta_{(t)}CO_2} \tag{6}$$

where $\Delta_{(t)}O_2$ and $\Delta_{(t)}CO_2$ are the changes of the $O_2$ and $CO_2$ mole fractions over time (tendencies) at a single height. Linear regression between $O_2$ and $CO_2$ can be applied and the slope gives the $ER_{atmos}$ value for a certain event or time period.

By assuming the mixed-layer theory, the tendencies in equation 6 depend on the surface and entrainment fluxes, together with the boundary layer height ($h$) (see Equation 1). Equation 6 can be rewritten by implementing Equation 1:

$$ER_{atmos} = -\frac{((F_{O_2})_s - (F_{O_2})_e)/h}{((F_{CO_2})_s - (F_{CO_2})_e)/h} \tag{7}$$

where $(F_{O_2})_s$ and $(F_{CO_2})_s$ are the net surface fluxes of $O_2$ and $CO_2$, and $(F_{O_2})_e$ and $(F_{CO_2})_e$ are the entrainment fluxes of $O_2$ and $CO_2$ respectively. For simplicity we ignored the advection term in Equation 1 here, and we will add it later (see Equation 9). As shown in Equation 2, the entrainment flux depends on the entrainment velocity ($w_e$) and the jump between the free troposphere and the boundary layer ($\Delta_{(ft-bl)}\phi$). Combining the definition for $ER_{forest}$ (Equation 5) with Equation 2, allows to rewrite Equation 7 to:

$$ER_{atmos} = ER_{forest} \cdot \left( \frac{1 + \frac{w_e \cdot \Delta_{(ft-bl)}O_2}{(F_{O_2})_s}}{1 + \frac{w_e \cdot \Delta_{(ft-bl)}CO_2}{(F_{CO_2})_s}} \right) = ER_{forest} \cdot \left( \frac{1 + \beta_{O_2}}{1 + \beta_{CO_2}} \right) \tag{8}$$

where $\Delta_{(ft-bl)}O_2$ and $\Delta_{(ft-bl)}CO_2$ are the jumps of $O_2$ and $CO_2$ between the free troposphere and the boundary layer, and $\beta_\phi$ is the ratio between the entrainment flux and the surface flux (Vilà-Guerau de Arellano et al., 2004). Equation 8 shows a clear relationship between $ER_{atmos}$ and $ER_{forest}$ by assuming the mixed-layer theory.

Using the definition of Equation 1, we can extend Equation 8 to include the effect of the other relevant large scale process, advection of $O_2$ ($adv_{O_2}$) and $CO_2$ ($adv_{CO_2}$):

$$ER_{atmos} = ER_{forest} \cdot \left( \frac{1 + \beta_{O_2} + \frac{h}{(F_{O_2})_s} \cdot adv_{O_2}}{1 + \beta_{CO_2} + \frac{h}{(F_{CO_2})_s} \cdot adv_{CO_2}} \right) \tag{9}$$

In Appendix A1 we analyse Equation 8 by determining when $ER_{atmos}$ would theoretically be close to $ER_{forest}$ during the day. We show that the $\beta$ values are specifically of importance here. When the $\beta$'s of $O_2$ and $CO_2$ are equal or very small, $ER_{atmos}$ gives the same signal as $ER_{forest}$. To fully unravel the diurnal variations of $ER_{atmos}$ under realistic conditions and identify influencing factors, we need to analyse a real case. Therefore, we study two observed situations by means of the coupled land-atmosphere model, CLASS which we will describe next.





## 3 Methods

In this section we describe the measurements that were used in this study, together with the mixed-layer model used to evaluate the $ER_{atmos}$ and $ER_{forest}$ signals.

### 3.1 Hyytiälä 2018 and 2019 measurement campaigns

The observational data were obtained from the SMEAR II Forestry Station of the University of Helsinki in Finland, located

in Hyytiälä, Finland (61° 51'N, 24° 17'E, +181 MSL) (Hari et al., 2013). The SMEAR II station serves as a measurement site within a boreal forest equipped with a 128 m tower for continuous measurements of atmospheric variables, fluxes and greenhouse gas mole fractions. These data are accessible at https://smear.avaa.csc.fi/. The tower is situated in a homogeneous Scots pine forest, with an average canopy height of 18 m and a podzolic soil. The measurement site is predominantly influenced by the surrounding forest and has minimal impact from signals of fossil fuel combustion (Faassen et al., 2023). For a

comprehensive description, see Hari et al. (2013).

During the spring/summer of 2018 (03-Jun until 02-Aug) and 2019 (10-Jun until 17-Jul), two measurement campaigns, referred to as OXHYYGEN (Oxygen in Hyytiälä), were conducted at Hyytiälä. Continuous measurements of both $O_2$ and $CO_2$ mole fractions were taken at two heights (125 m and 23 m). $O_2$ was measured using an Oxzilla II fuel cell analyser, and $CO_2$

was measured with a non-dispersive infrared (NDIR) photometer (URAS26). Further details about these measurements and the measurement system are given in Faassen et al. (2023). The measurement precision for $O_2$ was 19 per meg and for $CO_2$, it was 0.07 ppm. Although the precision for $O_2$ is relatively poor compared to previous studies, it is still adequate for studying the diurnal time scale, as shown in Faassen et al. (2023).

$O_2$ measurements are typically expressed as $\delta O_2/N_2$ ratios in 'per meg' units due to the high abundance of $O_2$ in the atmosphere (20.946%), classifying it as a non-trace gas. For direct comparison with $CO_2$ and implementation into our model, we convert per meg to ppm equivalents (ppmEq) by multiplying with the standard mole fraction of $O_2$ in air of 0.20946 (Keeling et al., 1998) since $O_2$ and $CO_2$ change concurrently. We use conserved variables, by using mole fractions to indicate the abundance of $O_2$ and $CO_2$ in the atmosphere, and therefore we can assume that the vertical profiles of $O_2$ and $CO_2$ are well-mixed

(Figure 2).

During the OXHYYGEN campaigns, radiosondes were launched on multiple days several times per day to quantify the impact of boundary-layer dynamics on the $O_2$ and $CO_2$ diurnal cycles. The radiosondes (Windsond, model S1H3-R, Sweden) measured vertical profiles of air pressure, wind speed, wind direction, relative humidity and temperature, with flight heights

reaching a maximum of 4500 m and rising rate of about 1.7 m s$^{-1}$. The measurements have an accuracy of 1.0 hPa for air pressure, 5% for wind speed, 0.2 C for temperature and 1.8% for the relative humidity. The temperature and humidity probe has



a response time of 6 seconds. For our analysis, we computed vertical profiles of potential temperature ($\theta$) and specific humidity ($q$) based on pressure, temperature and relative humidity measurements. Based on the vertical profile of vertical temperature, we also determine the boundary layer height with the parcel method. Figure 2 shows examples of vertical profile measurements

of $\theta$ for July 12, 2019, and July 24, 2018.

## 3.2  Modelling setup in CLASS

### 3.2.1  Implementation of $CO_2$ in CLASS

CLASS serves as a fundamental tool that enables further understanding of specific processes within the atmospheric boundary

layer. Several studies have shown that CLASS is successful in reproducing observational data (Vilà-Guerau de Arellano et al., 2012, 2019; Schulte et al., 2021). The study of Ouwersloot et al. (2012) specifically showed that CLASS is able to reproduce the boundary dynamics at the Hyytiälä measurement site. Within CLASS, the vegetation is described using a big-leaf model. The surface stomatal conductance that is representative for the canopy is up-scaled from leaf stomatal conductance by integrating over the leaf area index and incorporating soil moisture. The leaf stomatal conductance is calculated with the A-gs model. The

A-gs model relates leaf stomatal conductance ($g_s$) to the net leaf $CO_2$ assimilation (A) (Jacobs et al., 1996; Ronda et al., 2001). The model computes the dependece of $g_s$ and A with the internal $CO_2$ mole fraction, the amount of light, the atmospheric temperature, the vapor pressure deficit, and the soil water content at the root zone. Finally, the canopy net $CO_2$ assimilation is obtained with a function that is inspired by Fick's law of diffusion, that considers the difference of the atmospheric $CO_2$ and the internal $CO_2$ mole fractions, the aerodynamic resistance and the surface stomatal conductance. The soil respiration is

implemented as a function of soil temperature and soil moisture (Vilà-Guerau de Arellano et al., 2012). Combining the net assimilation ($A_n$) of the plants on canopy level and the soil respiration flux results in the net ecosystem exchange (NEE). This means that the model does not produces exactly the GPP and TER fluxes. These differences between $A_n$ and GPP, and soil respiration and TER are not directly relevant for our study and we therefore refer to GPP and TER in the following Sections, as these terms are more commonly used in the atmospheric $CO_2$ community. The water cycle is connected to the $CO_2$ cycle

through the surface stomata and the soil moisture inhibition functions for assimilation and respiration.

### 3.2.2  Implementation of $O_2$ in CLASS

To model both $ER_{forest}$ and $ER_{atmos}$, we incorporated the surface flux and the atmospheric mole fraction of $O_2$ into the CLASS model. We represent the surface flux of $O_2$ by multiplying the ER of assimilation ($ER_a$) and the ER of respiration ($ER_r$) with

the CLASS-calculated $CO_2$ fluxes at the canopy scale. We used the observationally derived $ER_a$ and $ER_r$ values as previously determined in Faassen et al. (2023) for the same site, which were 0.96 and 1.03 respectively. The net surface flux of $O_2$ was then resolved with the following equation:

$$F(O_2)_s = F_{CO_2(a)} \cdot -ER_a + F_{CO_2(r)} \cdot -ER_r \tag{10}$$





where $F(O_2)_s$ is the net $O_2$ surface flux above the canopy, $F_{CO_2(a)}$ is the net assimilation flux and $F_{CO_2(r)}$ is the soil respiration

flux. The change of atmospheric $O_2$ over time was resolved with Equation A1 (similar to Equation 1) and the entrainment

flux is based on Equation A2 (see also Equation 2). Note that the $ER_a$ from Faassen et al. (2023) was based on GPP fluxes

and this $ER_a$ is now linked to the net assimilation flux (GPP minus the photo and dark respiration) of the model (Jacobs

et al., 1996; Ronda et al., 2001). Seibt et al. (2004) and Ishidoya et al. (2013) showed that $ER_a$ values based on net assimilation

have similar values compared to the 0.96 based on GPP. We therefore expect that this discrepancy will not influence our results.


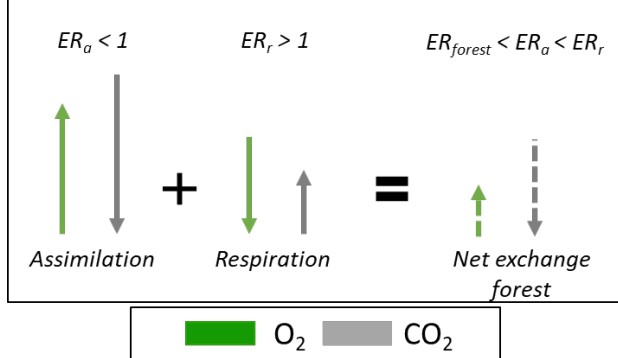

**Figure 3.** Schematic overview of how two processes with different ER signals produce a combined ER signal that is not necessarily the average of the two processes, nor necessarily falls inside the range of the two combined ER signals. This is due to the different signs for the $O_2$ and $CO_2$ fluxes. The example is given for combining the ER signal of Assimilation ($ER_a$) and Respiration ($ER_r$) into $ER_{forest}$ and uses values from our study that are by coincidence larger and smaller than 1.

It is important to note that the resulting $ER_{forest}$ signal is not the (weighted) average between $ER_a$ and $ER_r$, as was also

shown by Faassen et al. (2023). The $ER_{forest}$ signal results from the TER and GPP fluxes with different sizes and signs, with

each their own ER signals ($ER_r$ and $ER_a$ respectively). Figure 3 shows that the resulting $ER_{forest}$ signal does not necessarily

fall inside the range of the $ER_a$ and $ER_r$ signals because the TER and GPP have opposite signs of the $O_2$ and $CO_2$ fluxes. This

counter-intuitive situation can also occur for combining signals with different isotopic signatures (Miller and Tans, 2003).

### 3.2.3  Initial conditions

We determined initial and boundary conditions for two cases, to constrain the model to the observations. One case was based

on the year 2019 (base case) and the other case was based on the year 2018 (characterized by a warm summer in Finland;

Peters et al., 2020; Lindroth et al., 2020). Using the two years to initialize CLASS we were able to better constrain the veg-

etation's response in the CLASS model under extreme conditions. For each year, we selected one representative day for the

initialization and validation of the CLASS model. We used 10-07-2019 for the base case and an aggregate between 28-08-2018

and 29-08-2018 for the warm case. The final initial and boundary conditions for the initialisation of the CLASS runs can be





found in Tables C1 and C2 in the Appendix.

We deliberately made only minimal adjustments for the initialization of the 2018 case compared to the 2019 base case, to ensure consistency. We assumed that the initial relative humidity remained constant at 80%, regardless of temperature variations, similar to the studies of Vilà-Guerau de Arellano et al. (2012) and van Heerwaarden and Teuling (2014).

       We adjusted several parameters of the A-$g_s$ land surface scheme and the soil respiration to improve the agreement between
the surface fluxes of the model and the observations in Hyytiälä for both the base case (2019) and the warmer case (2018) (Table C1). We decreased the mesophyll conductance ($g_m$: 2 mm s$^{-1}$) to better match pine forest conditions (Gibelin et al., 2008; ECMWF IV, 2014; Visser et al., 2021). Furthermore, the reference temperature of $g_m$ ($T_{2(g_m)}$: 305 K) was increased to reduce afternoon plant stress and to make the CLASS run more comparable with the observations. Lastly, we adjusted the curvature of the drought response curve ($c_\beta$) from zero to 15% (Combe et al., 2015), considering that several studies demonstrate the pine
forest in Hyytiälä to be relatively resilient to lower soil moisture values and thus needing a higher ($c_\beta$) value (Gao et al., 2017; Lindroth et al., 2020).

### 3.2.4   Sensitivity analyses

We conducted two sensitivity analyses to gain a deeper understanding of the ER$_{atmos}$ behaviour under varying conditions and
to identify factors that lead to a smaller difference between ER$_{atmos}$ and ER$_{forest}$. With these sensitivity analyses, the effect of changing the different components of Equation 8 on ER$_{atmos}$ is tested. The first sensitivity analysis uses the 2019 base case, where we altered the initial jumps of $O_2$ and $CO_2$ to investigate the effect of background air with a different composition. By only changing the initial jump and keeping the rest of the 2019 case the same, we simulate situations in which the free troposphere mole fractions of $O_2$ and $CO_2$ have changed. In the second sensitivity analysis, we examined the impact of climate
conditions by modifying the soil moisture and air temperature, mimicking the conditions observed during the 2018 heatwave. Table A1 presents the variables used for initializing four cases for these two sensitivity studies.

## 4   Results

### 4.1   Validation of the $O_2$ and $CO_2$ model results

Overall, the modelled $O_2$ and $CO_2$ diurnal cycles match well with the observational data. Figures A3 and A2 in Appendix A4 show that CLASS accurately reproduces the diurnal cycles and captures the $O_2$ mole fraction changes on a daily time scale for both 2018 and 2019 (Figure A3b and A3c). The figure shows that the differences between the 2 years are relatively small and indicate that the boundary layer dynamics and the surface fluxes are well represented in CLASS. To accurately replicate the




rapid decrease of $CO_2$ and the sharp increase of $O_2$ during the rapid growth of the atmospheric boundary layer (between 6:30

and 11:30), we adjusted the jump between the boundary layer and the free troposphere ($\Delta_{(ft-bl)}$) for both $O_2$ (30 ppmEq) and

$CO_2$ (8 ppm), ensuring that the model aligned with the measurements. Based on values from previous studies, it is realistic

for the $CO_2$ jump to range between 8 ppm and 40 ppm (Vilà-Guerau de Arellano et al., 2004; Casso-Torralba et al., 2008).

However, there is limited data available to validate the jump of $O_2$, based on preliminary results from a campaign in Loobos,

the Netherlands, a jump of 30 ppmEq for $O_2$ seems reasonable. Our chosen combination of $O_2$ and $CO_2$ jumps remains an

uncertain component in our analysis and will be further discussed in Section 5.3.

## 4.2 Diurnal variability of $ER_{atmos}$ and $ER_{forest}$ in 2018 and 2019

In this section we discuss the diurnal variability of the $ER_{atmos}$ signals for both the 2018 and 2019 cases and use Equation 8

to explain this diurnal variability. First, we focus on the budget components (GPP, TER and entrainment) that influence the

tendencies of $O_2$ and $CO_2$ (Section 4.2.1). To complete the analysis, we support the numerical analysis with Equation 8 to gain

a more comprehensive understanding of the underlying processes driving the $ER_{atmos}$ signal for the 2019 case (Section 4.2.2).

### 4.2.1 The three distinct periods of the $ER_{atmos}$ signal during daytime

The $ER_{atmos}$ signals obtained for the 2018 and 2019 experiments display large variability throughout the daytime (panels a

and b in Figure 4). We identify three distinct periods during the day based on the processes shown in Figure 4d and 4e: 1)

the early morning regime (P1, 5:00-6:30 LT), characterised by an increasing net $CO_2$ flux out of the forest but a non-growing

boundary layer (Figure A3a), during which the $ER_{atmos}$ signal during P1 is still relatively close to $ER_{forest}$; 2) the entrainment

dominant period (P2, 6:30-11:30 LT), where air from a residual layer or air masses from the free troposphere are entrained

into the boundary layer and significantly influence the signals, leading to large $ER_{atmos}$ values, with an average greater than

3 and extreme values reaching close to 5; 3) the afternoon period (P3, 11:30-18:30 LT), where surface processes dominate

the observed signals and $ER_{atmos}$ moves slowly again towards $ER_{forest}$ and become more consistent with values expected for

surface processes. The $ER_{atmos}$ values during the three identified periods show a good agreement between the observations and

the model results (Table 1). This analysis confirms from a model perspective that values above 2 for $ER_{atmos}$, as we reported

in Faassen et al. (2023), are indeed possible. Figures 4d and 4e give first indications on what could cause these high values for

$ER_{atmos}$: high influence of entrainment and a different behaviour of the tendencies that influence $O_2$ compared to $CO_2$. In the

next Section we discuss the diurnal behaviour of $ER_{atmos}$ in more detail by using Equation 8.

We find that $ER_{forest}$ is much less variable throughout the day than $ER_{atmos}$ (Figure 4b and 4c). In the early morning and later

afternoon the $ER_{forest}$ value is lower than the mid-day period. This is caused by an almost equal TER flux (with a higher ER

signal) to the GPP flux (with a lower ER signal) caused by low sun light (Figure 3). During mid-day the assimilation of $CO_2$

by the canopy, with a lower ER signal, becomes increasingly dominant causing the $ER_{forest}$ signal to move closer to the $ER_a$





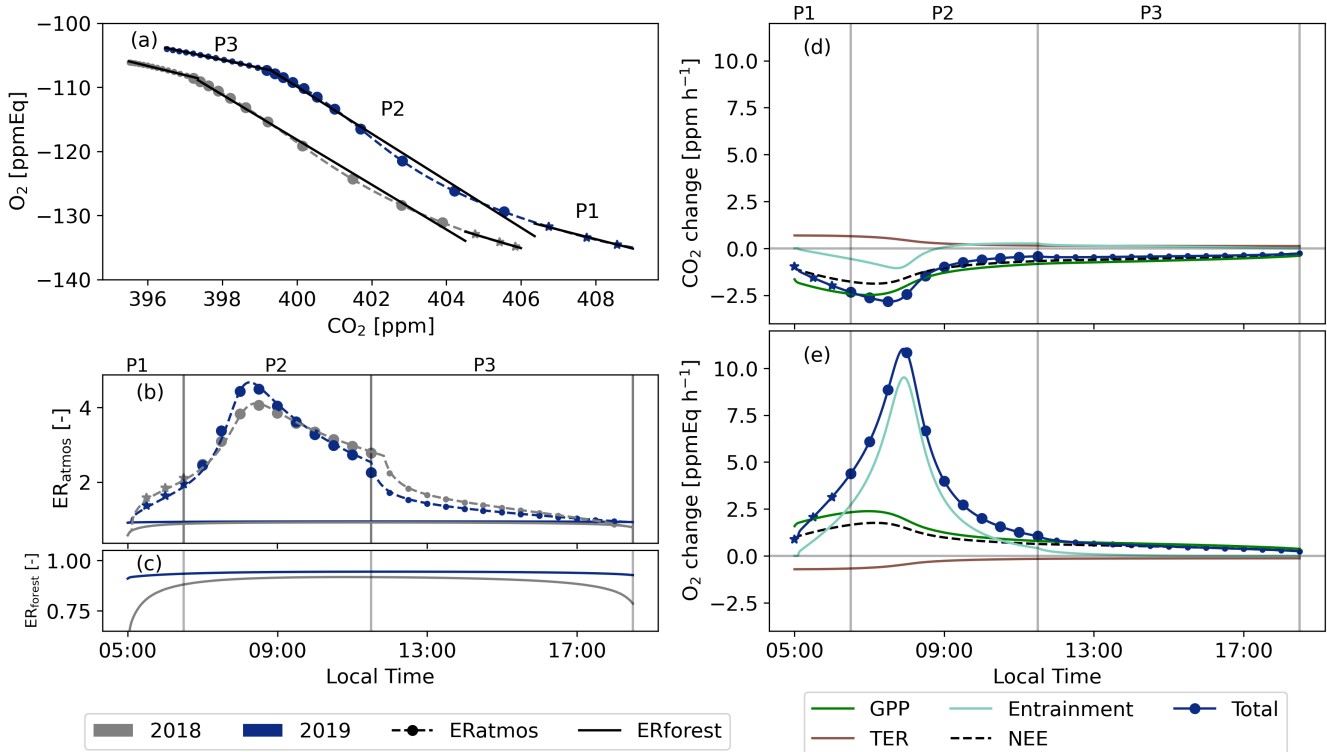

**Figure 4.** Diurnal cycles of $O_2$ and $CO_2$ mole fractions (a) and $ER_{atmos}$ and $ER_{forest}$ (b and c) as modelled with CLASS for the selected days in 2018 and 2019. We identify 3 distinct periods; P1 05:00-06:30 LT, P2 06:30-11:30 LT, and P3 11:30-18:30 LT, based on panels d and e, which show the tendencies for the 2019 case (change over time) for $CO_2$ and $O_2$ for each process that influences their mole fractions (Equation 1). The symbols represent half hourly averaged values of the CLASS model output.

value.

**Table 1.** $ER_{atmos}$ values (calculated as the slope of the $O_2$ and $CO_2$ mole fractions) and $ER_{forest}$ for the selected days in 2018 and 2019 for both observations (Obs) and the CLASS model for the three selected periods (P1: 5:00-06:30 LT, P2: 06:30-11:30 LT and P3: 11:30-19:30 LT). The uncertainties of the observed $ER_{atmos}$ and $ER_{forest}$ signals are determined following Faassen et al. (2023). Note that due to limited observational data we were unable to derive $ER_{atmos}$ values for P1 and P2 in 2018 and for P1 in 2019.

| | $ER_{atmos}$ (P1) | | $ER_{atmos}$ (P2) | | $ER_{atmos}$ (P3) | | $ER_{forest}$ (P1-P3) | |
|---|---|---|---|---|---|---|---|---|
| Year | Obs | Model | Obs | Model | Obs | Model | Obs | Model |
| 2018 | n.a. | 1.72 | n.a. | 3.50 | $1.67 \pm 0.51$ | 1.43 | $0.87 \pm 0.07$ | 0.90 |
| 2019 | n.a. | 1.48 | $3.33 \pm 0.31$ | 3.66 | $1.23 \pm 0.10$ | 1.24 | $0.86 \pm 0.06$ | 0.94 |





### 4.2.2 Explanation of the large $ER_{atmos}$ values

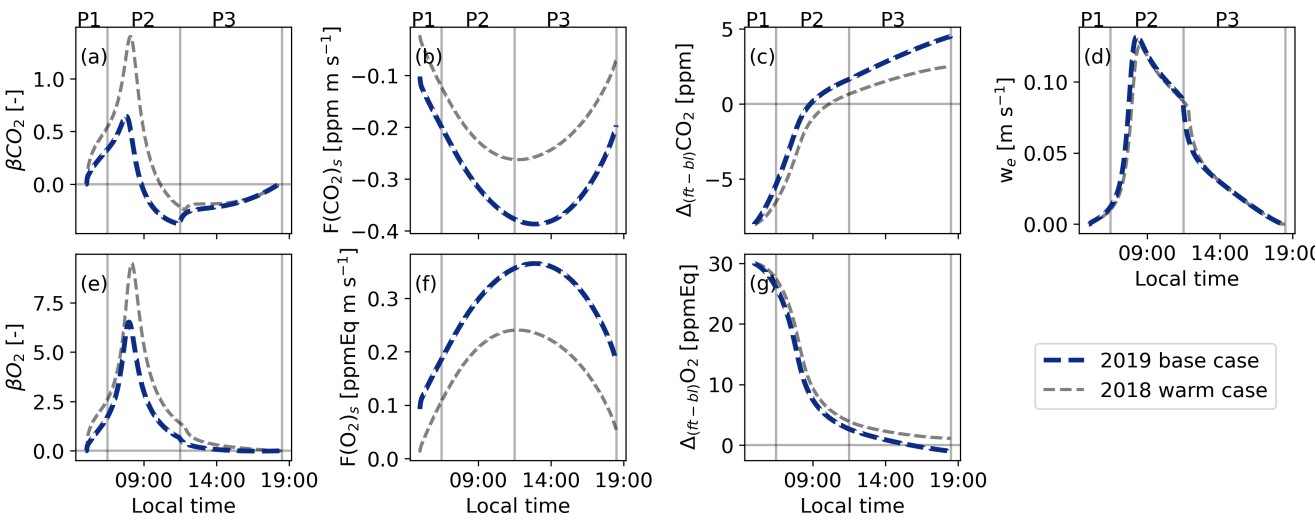

**Figure 5.** The diurnal variability of the different components of Equation 8 for the base case (2019) and the warm case (2018) derived with the CLASS model. (a) and (e) show the $\beta$ values for $CO_2$ and $O_2$ where $\beta$ is the entrainment flux divided by the surface flux (Equation 8), (b) and (f) show the net surface flux, (c) and (g) show the jumps between the free troposphere and the boundary layer ($\Delta_{(ft-bl)}$), (d) shows the entrainment velocity ($w_e$) and (e) shows the resulting ratio between $ER_{atmos}$ and $ER_{forest}$ during the day. The vertical lines represent three distinct periods: 05:00-06:30 LT (P1), 06:30-11:30 LT (P2), 11:30-18:30 LT (P3).

Analysing the diurnal cycle of the different components of Equation 8 for the 2019 case reveals that the peak value of $ER_{atmos}$
during P2 is caused by the higher values for $\beta_{O_2}$ compared to $\beta_{CO_2}$ (Figure 5). The difference between $\beta_{O_2}$ and $\beta_{CO_2}$ is a result of a high $\Delta_{(ft-bl)}O_2/\Delta_{(ft-bl)}CO_2$ ratio (higher than 3). These terms $\Delta_{(ft-bl)}O_2$ and $\Delta_{(ft-bl)}CO_2$ represent the jump across the boundary layer top, and each has a different diurnal cycle caused by a different surface flux (Figure 5c and 5g). These different diurnal cycles for the jumps lead to an increase in the $\Delta_{(ft-bl)}O_2/\Delta_{(ft-bl)}CO_2$ ratio, consequently raising the ratio between the $\beta$ values. This effect is further amplified by a higher surface flux of $CO_2$ compared to $O_2$, caused by an $ER_{forest}$ value that is
slightly lower than 1. The peak value of $ER_{atmos}$ during P2 occurs when both $w_e$ and the $\Delta_{(ft-bl)}O_2/\Delta_{(ft-bl)}CO_2$ ratio are high and the surface fluxes are still relatively low. This combination contributes to the distinctive peak in $ER_{atmos}$ observed during P2.

Later in the afternoon (P3), both $\beta$ values gradually decrease and become similar, resulting in an $ER_{atmos}$ signal that becomes closer to $ER_{forest}$. This indicates that $ER_{atmos}$ becomes more representative for surface processes (see also Sect. A1). This de-
crease in P3 is primarily caused by a reduction in the entrainment velocity ($w_e$) (Figure 5d), indicating a slow growth of the atmospheric boundary layer at end of the day (Figure A3). Additionally, the $\beta$ values become more similar because $\Delta_{(ft-bl)}O_2$ moves closer to $\Delta_{(ft-bl)}CO_2$ during this period (Figure 5c and 5g), caused by the mixing of air with the surface.



The $ER_{atmos}$ signals exhibit higher values compared to the theoretical analysis in Sect. A1 because the diurnal cycles of the
components of Equation 8 are taken into account (Figure A1 vs Figure 5). Each component of Equation 8 follows its individual
diurnal cycle, leading to higher $ER_{atmos}$ values. Consequently, $ER_{atmos}$ is integrating individual contributions of several pro-
cesses, particularly during P2, since it is dominated by the influence of mixing with large scale processes. Careful consideration
is needed when interpreting the $ER_{atmos}$ signal during this period. During P3, the $ER_{atmos}$ signal appears to align with $ER_{forest}$
at the end of the day. However, in the 2019 case, this alignment was only observed for a very short period.


We find only small differences for the diurnal behaviour of the $ER_{forest}$ and $ER_{atmos}$ signal between the 2018 and 2019 case
(Figure 4 and Figure 5). The $ER_{forest}$ value is lower in 2018 compared to 2019. This can be attributed to a higher respiration
flux caused by the elevated temperatures during that day which also results in an increase of the soil temperature (Figure
A3e). While we do not have direct measurements of $ER_r$ and $ER_a$ for both 2018 and 2019, it is likely that the overall diurnal
cycle pattern of $ER_{forest}$ in Figure 4 (low $ER_{forest}$ values in the morning and afternoon, higher $ER_{forest}$ values during mid-day)
for both years would have remained consistent. Previous studies suggest that $ER_r$ is generally higher than $ER_a$, even under
different atmospheric conditions (Angert et al., 2015; Fischer et al., 2015; Hilman et al., 2022). The effect of a warmer and
dryer environment on the $ER_{atmos}$ signal will be further quantified in Sect. 4.3.2 with a more extreme case.

### 4.3  Sensitivity analyses: effects of changing large scale conditions

With the next two sensitivity analyses we evaluate whether the 2019 case was an exception and if there are cases where the
$ER_{atmos}$ signal could become equal to $ER_{forest}$ when large scale conditions would change. We focus on the effect of changes in
the background air (Sect. 4.3.1) and the effect of climate (changes in soil moisture and air temperature) (Sect. 4.3.2). Figure
6 shows how $ER_{atmos}$ is formed by the different components of Equation 8, and how the variables changed in the sensitivity
analyses impact these components and therefore $ER_{atmos}$.


#### 4.3.1  Effects of changing background air on $ER_{atmos}$

Changing the background air in the free troposphere by decreasing the initial jump ratio or the jump sizes of $O_2$ and $CO_2$
compared to the 2019 case, moves the $ER_{atmos}$ signal closer to $ER_{forest}$ during P2 and P3 (Figure B1). A lower jump ratio than
the 2019 case, but still relatively high jump values ($\Delta_{(ft-bl)}O_2 = 30$ and $\Delta_{(ft-bl)}CO_2 = -20$) lead to a decrease in the peak of
$ER_{atmos}$ during P2 and bring $ER_{atmos}$ closer to $ER_{forest}$ during P3 (yellow line in Figure B1). As the jump ratio decreases, the
$\beta_{O_2}$ becomes less dominant and more identical to $\beta_{CO_2}$. When the $O_2$ and $CO_2$ $\beta$ values become closer, the $ER_{atmos}$ value also
moves closer to $ER_{forest}$ (Figure 6). However, this does not necessarily mean that the surface has become more dominant since
the $\Delta_{(ft-bl)}$ values are still relatively high.

Reducing the jump sizes of both $O_2$ and $CO_2$ ($\Delta_{(ft-bl)}O_2 = 10$ and $\Delta_{(ft-bl)}CO_2 = -8$) still results in a relatively high peak
for $ER_{atmos}$ during P2 and bring $ER_{atmos}$ closer to $ER_{forest}$ during P3 (purple line in Figure B1). Including the diurnal cycle



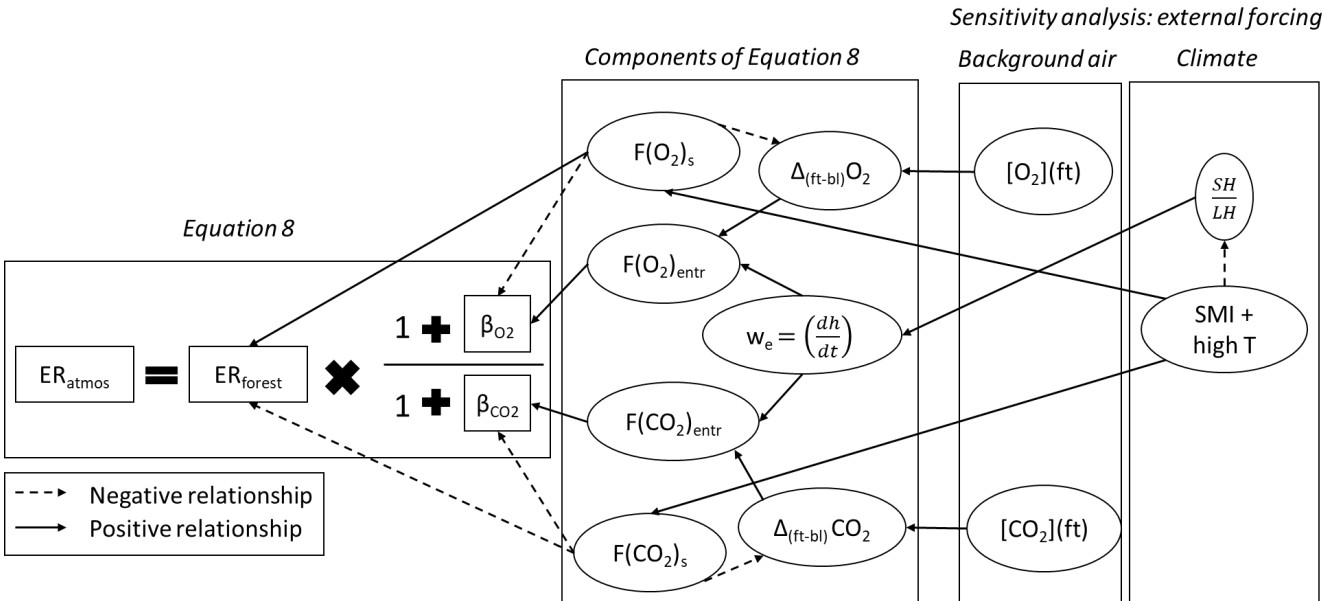

**Figure 6.** The components of Equation 8 and how these influence the $ER_{atmos}$ signal, including: the exchange ratio of the forest ($ER_{forest}$), the ratio between the net surface flux ($F_s$) and the entrainment flux ($F_{entr}$) which result in the $\beta$, the jump between the free troposphere and the boundary layer ($\Delta_{(ft-bl)}$) and the entrainment velocity ($w_e$). The right part of the Figure shows the variables that are changed in the two sensitivity analyses: the background air in the free troposphere ($[O_2](ft)$ and $[CO_2](ft)$) and the initial Soil Moisture Index (SMI) in combination with a high initial potential temperature ($\theta_0$) that will influence the ratio between the sensible heat flux (SH) and the latent heat flux (LH) at the surface. The dotted arrows indicate a negative influence and the solid arrows indicate a positive influence.

of the jumps accounts for the effect that the $CO_2$ jump changes from a negative to a positive value during the day. When the initial $CO_2$ jump is lower, the sign change occurs earlier in the day and leads to a more negative $\beta_{CO_2}$ value. This leads to higher $ER_{atmos}$ values during P2 (Figure 6). In contrast, a lower jump size would cause the $ER_{atmos}$ signal to move more quickly

towards $ER_{forest}$ during P3 because the surface fluxes dominate over the lowered entrainment flux.

Guided by our theoretical and numerical results and constrained by observations, a high $ER_{atmos}$ signal during the entrainment dominant period (P2) can therefore be a result of two cases:

1. The $\Delta_{(ft-bl)}O_2$ is substantially larger compared to $\Delta_{(ft-bl)}CO_2$ and therefore $\beta_{O_2}$ dominates over $\beta_{CO_2}$.

2. $\Delta_{(ft-bl)}CO_2$ changes sign from negative to positive and as a result $\beta_{CO_2}$ becomes negative resulting in a denominator closer to zero.

Changes in the background air result in a distinct change in the diurnal pattern of $ER_{atmos}$. The difference between the $ER_{atmos}$ and $ER_{forest}$ signal could therefore provide extra information on the changes of large scale processes. This is further discussed






### 4.3.2 Effect of climate conditions on $ER_{atmos}$ and $ER_{forest}$

By studying the influence of changes in air temperature and soil moisture on the $ER_{atmos}$ signal (see Figure 7), we gain insights into how climate (spring versus summer or heat wave versus normal conditions) can effect $ER_{atmos}$ compared to $ER_{forest}$ and this allows us to study the effects of future climate with dryer and warmer conditions. The 2018 case already showed how the

$ER_{atmos}$ signal could change with a decreasing soil moisture and increasing temperature compared to a more normal year in 2019 (Figure 4 and 5). As a next step, we evaluate the full range of how $ER_{atmos}$ could change and how $ER_{atmos}$ compares to $ER_{forest}$. Given the same net radiation, higher soil moisture levels enhance soil respiration, photosynthesis and latent heat fluxes, and thus decrease the sensible heat flux because of the energy balance closure (see Figure 6). A lower sensible heat flux would decrease the boundary layer growth and as a result decrease the entrainment velocity. In addition, higher air temperatures ac-

celerate both the photosynthesis and the respiration until a threshold (Jacobs et al., 1996), resulting in increased GPP and TER fluxes. Lower soil moisture levels in combination with higher temperatures can stress plants, leading to decreased $O_2$ and $CO_2$ surface fluxes and enhanced the sensible heat flux. Thereby increasing the boundary layer growth and the entrainment velocity (Equation 2 and 4). Note that there are also minor changes for $ER_{forest}$ when the soil moisture index (SMI: [soil moisture - $w_{wilt}$]/[$w_{fc}$ - $w_{wilt}$]) and air temperature change as a result of GPP and TER changes.


Increasing or decreasing the SMI in combination with changes in air temperature makes the diurnal variability of $ER_{atmos}$ more complex because all the components of Equation 8 are now affected (Figure 6 and Figure 7). We focus on two situations of Figure 7: a low soil moisture (red symbol) and a high soil moisture case (green symbol), both with higher temperatures compared to the 2019 case (Figure B2). A lower soil moisture of 0.14 $m^3$ $m^{-3}$ (SMI = 0.27) with an air temperature of 290 K

decreases the $ER_{atmos}$ signal during P2 and increases the $ER_{atmos}$ signal during P3 compared to the 2019 base case (the red lines in Figure B2 and red symbol in Figure 7). The lower $ER_{atmos}$ values during P2 are primarily a consequence of a more dominant entrainment flux. Due to a decrease in the $O_2$ and $CO_2$ surface fluxes because of stressed plants, both the $\Delta_{(ft-bl)}$ values for $O_2$ and $CO_2$ change relatively slower and remain high. Higher $\Delta_{(ft-bl)}$ values, along with a higher entrainment velocity caused by a higher sensible heat flux, lead to elevated entrainment fluxes. By increasing both the $O_2$ and $CO_2$ entrainment fluxes and

decreasing the $O_2$ and $CO_2$ net surface fluxes, the $\beta$ values increase and the ratios of the $\beta$ values move towards the $\Delta_{(ft-bl)}$ ratios. As a result the $ER_{atmos}$ also moves towards the $\Delta_{(ft-bl)}$ ratios multiplied with the $ER_{forest}$ signal (Figure 6). This is similar with the effect observed when increasing both the initial jumps of $O_2$ and $CO_2$ (Sect. 4.3.1). The $\beta$ values stay high during P3 because of the low net $O_2$ and $CO_2$ surface fluxes. Therefore, the $ER_{atmos}$ signal also remains close to the ratio of the $\Delta_{(ft-bl)}$ values during P3 and the $ER_{atmos}$ signal does not approach $ER_{forest}$ (Figure 6).


In contrast, a higher soil moisture of 0.22 $m^3$ $m^{-3}$ (SMI = 0.64) with an air temperature of 290 K increases the $ER_{atmos}$ signal during P2 and decreases the $ER_{atmos}$ signal during P3 compared to the 2019 base case (the green lines in Figure B2 and



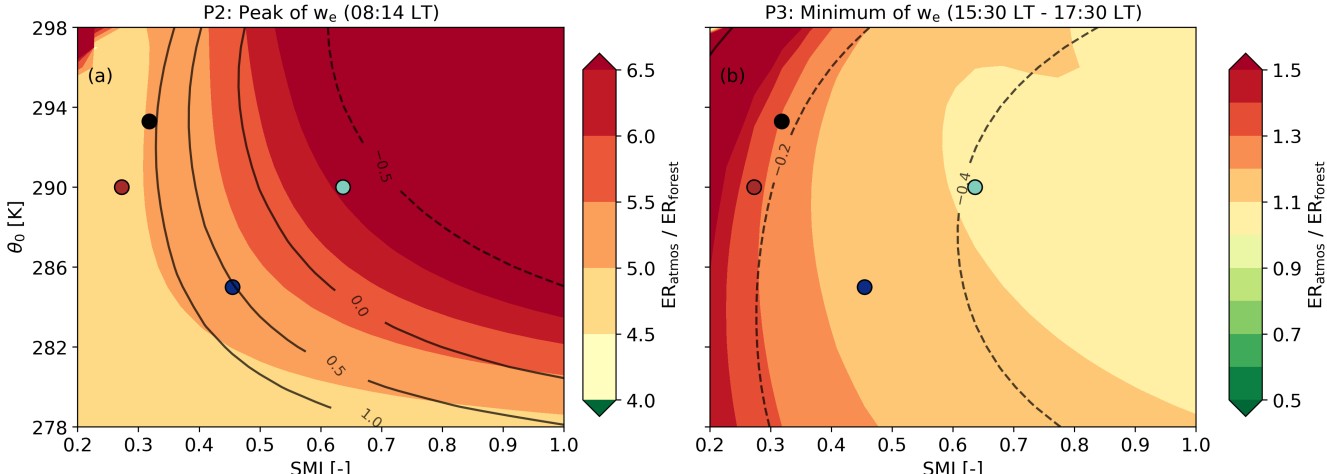

**Figure 7.** Evaluation of the ratio between $ER_{atmos}$ and $ER_{forest}$ as a function of two key variables that show the effect of a drier and warmer climate: the Soil Moisture Index (SMI) and the initial potential temperature ($\theta_0$). Two moments in the day are analysed, (a) during the maximum value of $w_e$ at 08:14 LT (P2) and (b) at the end of the day between 15:30 - 17:30 LT when the $w_e$ is minimal (P3). The grey lines in (a) indicate $\beta_{CO_2}$ values, which is the ratio between the entrainment and the surface flux. The grey lines in (b) indicate net $CO_2$ surface flux values in ppm m s$^{-1}$. The coloured symbol (brown and light blue) indicate the example cases that are also shown in Figure B2 and the black dot is the 2018 case and the dark blue dot the 2019 case.

green symbol in Figure 7). This is consistent with the effect observed when lowering the initial $\Delta_{(ft-bl)}$ value (Sect. 4.3.1).

In addition to the conclusions in Sect. 4.3.1 on the causes of the high $ER_{atmos}$ signals during P2, the sensitivity analyses for changing climate conditions showed that the large differences between $ER_{atmos}$ and $ER_{forest}$ at the end of the day (P3) can be caused by:

1. A substantially larger $\Delta_{(ft-bl)}O_2$ compared to $\Delta_{(ft-bl)}CO_2$ causing $\beta_{O_2}$ to dominate over $\beta_{CO_2}$.

2. High $\beta_{O_2}$ and $\beta_{CO_2}$ values because of high $O_2$ and $CO_2$ entrainment fluxes and/or low net $O_2$ and $CO_2$ surface fluxes.

Our two sensitivity analyses show that several factors, including the entrainment velocity, the $\Delta_{(ft-bl)}$ values and their ratio and the net surface flux of $CO_2$ can significantly influence the diurnal behaviour of $ER_{atmos}$. When using $ER_{atmos}$ as an indication of $ER_{forest}$, these four factors should be carefully considered. This is crucial to correctly interpret $ER_{atmos}$ values and to understand the underlying processes that influence the carbon exchange above a forest canopy.




## 5 Discussion

### 5.1 Evaluation of the CLASS model

Our implementation of $O_2$ in the CLASS model could be improved in future studies. Similar to the approach used by Yan et al. (2023), both the $ER_r$ and $ER_a$ signals were kept constant and did not account for potential variations under different climate conditions. To advance our understanding of the ER signals over forest canopies, it is crucial to incorporate ER signals that can respond to varying soil and atmospheric conditions. For instance, the $ER_r$ of the soil respiration depends on air temperature and soil moisture (Hilman et al., 2022; Angert et al., 2015), while the $ER_a$ is primarily influenced by nitrogen content and light on leaf level (Bloom, 2015; Fischer et al., 2015). Additionally, in our current implementation, we did not include the ER for stem respiration ($ER_{stem}$) (Hilman and Angert, 2016) due to the absence of stem respiration in the CLASS model.

While we utilized CLASS in this study as a proof of concept to demonstrate how $ER_{atmos}$ changes during the day, employing a more elaborate model could allow for more detailed exploration of these $ER_{atmos}$ dynamics and the contributions of various processes. Models with more vertical levels could simulate vertical gradients and analyze differences in the $ER_{atmos}$ signal at various heights, similar to the approach in Yan et al. (2023). Implementing more vertical levels gives the opportunity to determine the dominance of large scale processes over small scale surface processes at different measurement heights. By incorporating a canopy into the model, the surface resistance could be accounted for, enhancing the accuracy of the modeled surface fluxes. Furthermore, exploring larger temporal and spatial scales could yield valuable insights in the variability of $ER_{forest}$ over time and space. Increasing the temporal scales gives the opportunity to improve estimates of the $ER_{forest}$ values used as the globally used 1.1 biosphere ER signal (Severinghaus, 1995).

### 5.2 How $ER_{atmos}$ should be used

Single height $O_2$ and $CO_2$ measurements and their $ER_{atmos}$ signal should be analysed very carefully when using it as an indicator for surface exchange. During the complete diurnal cycle, $ER_{forest}$ should be utilized as the primary indicator of the ER signals from the surface, while $ER_{atmos}$ should not be used for this purpose. In situations where only one height measurement is available, and therefore only $ER_{atmos}$ can be obtained, a first estimate of $ER_{forest}$ could be made using $ER_{atmos}$. The $ER_{atmos}$ signal at the end of the day should then be used to avoid the large influence of entrainment earlier in the day. However, any analysis or discussion based on this estimation should include a comprehensive examination of how entrainment might have influenced the $ER_{atmos}$ signal.

Several studies have showed that $ER_{atmos}$ can also serve as an indicator of potential advection from carbon source/sink regions (Ishidoya et al., 2020, 2022a). Nevertheless, caution should be exercised when directly inferring the specific source based solely on the $ER_{atmos}$ value. Equation 9 shows that mixing advected air with the air above a forest will result in an $ER_{atmos}$ signal that cannot be directly linked to the source of the advected air. Next to the influence of the surface and entrainment, the $ER_{atmos}$





signal also depends on the magnitude of the advected flux because of the effect that mixing two ER signals with opposite fluxes does not result in a weighted average (Figure 3). Advection of a source with the same ER signal but with different magnitudes

can therefore give different $ER_{atmos}$ signals. A solution could be to include other pollutants in the analysis such as $NO_x$ or CO (Liu et al., 2023a).

When two or more measurement heights of $O_2$ and $CO_2$ are available, and therefore $ER_{forest}$ can be derived, $ER_{atmos}$ of a single height could be used to provide extra information on large scale processes, by analysing the difference between $ER_{atmos}$

and the $ER_{forest}$ signal. Throughout the day, $ER_{atmos}$ provides insights into larger scale processes, while $ER_{forest}$ reflects local or small-scale processes. Therefore, any discrepancy between $ER_{atmos}$ and $ER_{forest}$ indicates a significant influence of large scale processes. Nonetheless, the exact difference between $ER_{atmos}$ from $ER_{forest}$ should not be used as an indication of the strength of the influence of large processes. To get more detail on how the large scale processes change between days, the diurnal cycle of $ER_{atmos}$ has to be compared during the entrainment dominant period (P2) and the surface dominant periods (P3). During

P2, an increase in the difference between $ER_{atmos}$ and $ER_{forest}$ can be attributed to either a low $\beta_{CO_2}$ or a change in the jump $(\Delta_{(ft-bl)})$ ratio. When a low $\beta_{CO_2}$ causes the high $ER_{atmos}$ values during P2, the $ER_{atmos}$ signal during P3 should be closer to $ER_{forest}$, compared to the situation where a high jump ratio leads to elevated $ER_{atmos}$ values.

## 5.3 Different $\Delta_{(ft-bl)}$ ratios

Knowing the vertical profile of $O_2$ and $CO_2$ especially during sunrise is essential to gain a more comprehensive understanding of the formation of different jump ratios ($\Delta_{(ft-bl)}O_2$ / $\Delta_{(ft-bl)}CO_2$) and to better interpret the diurnal behavior of the $ER_{atmos}$ signal. However, due to lack of observational data we cannot validate the vertical profile of $O_2$ and $CO_2$ and the jump ratios and we recommend for future measurement campaigns to include vertical measurements of both species, e.g. by flask sampling from aircraft. Although some studies have measured vertical profiles of $O_2$ and $CO_2$, they primarily focused on well-mixed

profiles or profiles over the ocean (Morgan et al., 2019; Stephens et al., 2021; Ishidoya et al., 2022b). Hence, careful consideration of the timing and location of the vertical measurements is important to advance our knowledge of the diurnal behaviour of $ER_{atmos}$.

Due to lack of observational data, we show with hypothetical situations that various jump ratios become possible (Figure 8).

Both the $O_2$ and $CO_2$ jumps are formed as a result of three processes; the mixed-layer value before sunset (2a), the surface flux during the night (1) and the free troposphere value with the lapse rate (3) (we assume the lapse rate to be 0 mol m$^{-1}$ for $CO_2$ and $O_2$). Most cases indicate that $\Delta_{(ft-bl)}O_2$ is larger than $\Delta_{(ft-bl)}CO_2$ above a forest, primarily because $ER_{forest}$ is higher than 1.0 during the night ($ER_r > 1.0$). It is noteworthy that the movement of the mixed-layer values from (P2a) to (P2b) in Figure 8 differs from its depiction in Figure 2c, where the focus was primarily on the transition between sunrise and sunset.

We ignore the effect of subsidence on the jump evaluation in this analysis, caused by mesoscale or synoptic processes, because





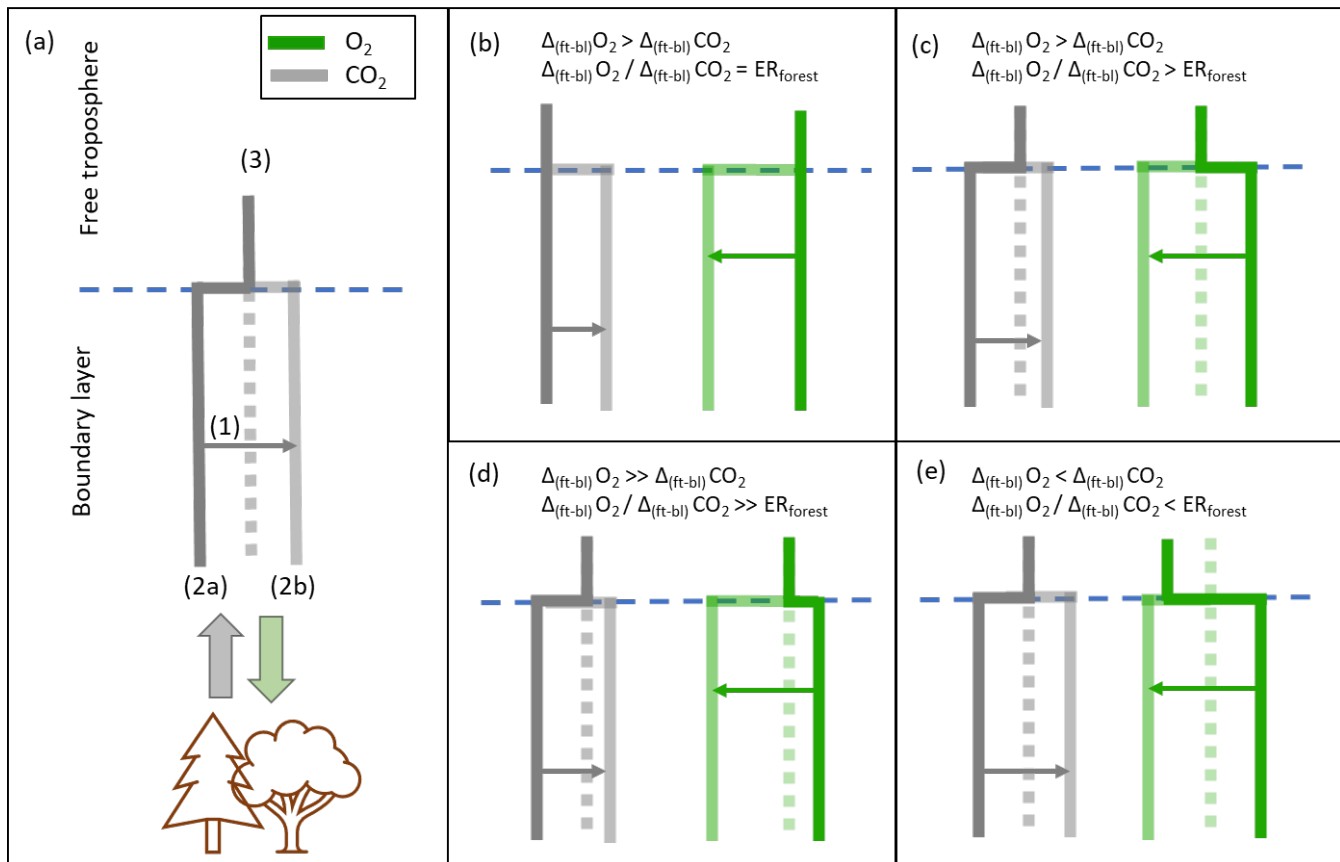

**Figure 8.** A schematic overview of how different jump ratios are possible between $O_2$ ($\Delta_{(ft-bl)}O_2$) and $CO_2$ ($\Delta_{(ft-bl)}CO_2$) and how the ratio relates to the Exchange Ratio of the forest ($ER_{forest}$). (a) shows an overview and the other panels show different possibilities of different jump ratios: (b) $\Delta_{(ft-bl)}O_2/\Delta_{(ft-bl)}CO_2$ is equal to $ER_{forest}$, (c) $\Delta_{(ft-bl)}O_2/\Delta_{(ft-bl)}CO_2$ is larger than $ER_{forest}$, (d) $\Delta_{(ft-bl)}O_2/\Delta_{(ft-bl)}CO_2$ is much larger than $ER_{forest}$, (e) $\Delta_{(ft-bl)}O_2$ is smaller than $\Delta_{(ft-bl)}CO_2$. The numbers in (a) indicate: (1) Surface flux size during night, (2a) mixed layer mole fraction just before sunset, (2b) mixed layer mole fraction just before sunrise and (3) mole fraction in the free troposphere with the corresponding lapse rate, which for $O_2$ and $CO_2$ the lapse rate is assumed to be 0 ppm m$^{-1}$.

it is likely of less importance compared to the other three processes.

It is highly likely that the jump ratio between $O_2$ and $CO_2$ cannot be directly linked to a specific ER for a certain process because of the interplay between the three processes that form the $O_2$ and a $CO_2$ jump (Figure 8d). The likelihood of both 490 $\Delta_{(ft-bl)}O_2$ and $\Delta_{(ft-bl)}CO_2$ being zero at the end of the day is low because the surface flux during the day would form a jump (Figure 8c). Additionally, it is possible that the $\Delta_{(ft-bl)}O_2$ is smaller than $\Delta_{(ft-bl)}CO_2$ at the end of the day due to the daytime $ER_{forest}$ being smaller than 1.0 (Figure 8d). Consequently, $O_2$ will exhibit a faster movement across the zero line,




resulting in a significantly larger $\Delta_{(ft-bl)}O_2/\Delta_{(ft-bl)}CO_2$ ratio compared to $ER_r$.

Decoupling between the free troposphere and the boundary layer can lead to a scenario in which $\Delta_{(ft-bl)}CO_2$ becomes larger than $\Delta_{(ft-bl)}O_2$ (Figure 8e). This can e.g. occur when the influence of fossil fuel sources causes a decrease in the $O_2$ mole fraction and an increase in the $CO_2$ mole fraction in the free troposphere, but large surface fluxes from the forest prevent such changes from occurring in the boundary layer. The jump ratio in this case again cannot be attributed to a single process. Some studies have demonstrated that decoupling between the boundary layer and the free troposphere can occur, leading to
different ER signals (Sturm et al., 2005; van der Laan et al., 2014).

## 5.4    Comparison with other studies

To the best of our knowledge, no previous studies have reported such high deviations of $ER_{atmos}$ from $ER_{forest}$, or $ER_{atmos}$ values higher than 2 for above forest canopy measurements as we found in Faassen et al. (2023). Only Liu et al. (2023b) found a dif-
ficult to explain nonlinear relationship between $O_2$ and other pollutants. While some differences between $ER_{atmos}$ and $ER_{forest}$ have been observed in previous studies, these differences typically fall within a range of 0.5 (Seibt et al., 2004; Ishidoya et al., 2015; Battle et al., 2019; Yan et al., 2023). A possible reason for these smaller differences could be that most studies do not focus on such detailed diurnal analyses of $ER_{atmos}$ for specific days but rather aggregate data from multiple days, which could mitigate the extreme effects of entrainment by combining various jump possibilities. However, even in the study by Stephens
et al. (2007), in which measurements at different heights are shown, no discernible difference in the $ER_{atmos}$ signal for various diurnal cycles was observed, a finding that contrasts with our own analysis. Large values for $ER_{atmos}$ have only been found at high latitude measurement stations (Sturm et al., 2005), due to the influence of the ocean.

There are several possibilities that could explain a constant $ER_{atmos}$ signal during the day, which are not shown in our study.
One possibility is that entrainment dominates throughout the day, caused by high jumps. If both the $O_2$ and $CO_2$ jumps are extremely high while the surface flux remains low, the $ER_{atmos}$ value reflects the ratios between the jumps. In this scenario, $ER_{atmos}$ cannot be used as an accurate indicator for the surface processes. Another explanation could be that the $ER_{forest}$ signal is exactly 1.0 and entrainment is relatively low. When $ER_{forest}$ equals 1.0, the diurnal cycle of the jumps would respond similarly. Together with a low entrainment flux (resulting from low jumps), it could lead to a constant $ER_{atmos}$ signal. Additionally, when
the peak of $ER_{atmos}$ occurs rapidly, there is a possibility that a low measurement precision would miss the extreme changes of $ER_{atmos}$. However, even in such cases, $ER_{atmos}$ would still be influenced by entrainment, although its impact may be less discernible. It is crucial to note that, in all these cases, $ER_{atmos}$ remains influenced by entrainment to varying degrees.

Our study provides evidence that $ER_{atmos}$ is almost always influenced by large scale processes and their diurnal variability,
specifically entrainment, making it important to exercise caution when using it as an indicator for the surface ER processes. Instances where $ER_{atmos}$ remains constant throughout daytime and serves as a reliable indication for $ER_{forest}$ are rare. In com-





parison to previous studies (Seibt et al., 2004; Stephens et al., 2007; Ishidoya et al., 2013; Battle et al., 2019), it is unclear why Faassen et al. (2023) yields such extreme values for $ER_{atmos}$ while the other studies do not show this, even though our modelling study here confirms the extreme $ER_{atmos}$ values. Therefore, we recommend conducting more studies or performing detailed analyses of existing $O_2$ and $CO_2$ data sets to gain a better understanding of how changes in $ER_{atmos}$ vary with time and space.

### 5.5 Comparison with other multi-tracer analyses

The impact of changes in large scale conditions such as entrainment on multi-tracer analyses above forest canopies extends beyond atmospheric $O_2$, encompassing other carbon cycle tracers such as carbon and oxygen isotopes ($\delta^{13}C$ and $\delta^{18}O$) (Wehr et al., 2016), and carbonyl sulfide (COS) (Whelan et al., 2018). Caution is required when employing methods of determining ratios between two species (eg. leaf relative uptake for COS and the ratios between different isotopes) that rely solely on single-height measurements. However, the influence of entrainment on these ratios would be less extreme compared to the $ER_{atmos}$ signal because both COS and isotopes move in the same direction as $CO_2$ itself. This is different compared to $O_2$, which always moves in the opposite direction compared to $CO_2$. When both species that form the ratio move in the same direction, ratios of different processes could be averaged and a one height measurement is more readily interpretable. Nevertheless, entrainment would still cause the two compounds that form the ratio to behave differently. We therefore emphasize the need to separately analyze the composition of the signal for each compound when ratios are analyzed.

Furthermore, we demonstrate in this study the potential of using $ER_{atmos}$ as an indicator of the extent of large scale processes. Additional tracers can contribute to this question. $\delta^{13}C$, $\delta^{18}O$ and COS signals exhibit differences between the surface and the free troposphere. Similar to $O_2$, the onset of entrainment causes these signals to mix, yielding insights into how large scale processes influence the carbon cycle above a canopy (Berkelhammer et al., 2014; Vilà-Guerau de Arellano et al., 2019). By combining various tracers for $CO_2$, we can create a comprehensive picture of the effects of small scale and large scale processes that influence carbon exchange.

### 6 Conclusions

We used a mixed-layer model to analyze the diurnal behavior of two Exchange Ratio (ER = $O_2$/$CO_2$) signals above a forest canopy: the ER of the atmosphere ($ER_{atmos}$ determined from $O_2$ and $CO_2$ mole fraction measurements at a single height above the canopy) and the ER of the forest ($ER_{forest}$ determined from $O_2$ and $CO_2$ fluxes derived from the vertical gradient observations at two levels). We disentangled the biophysical processes influencing $ER_{atmos}$ to interpret single height $O_2$ and $CO_2$ measurements and to evaluate how both $ER_{atmos}$ and $ER_{forest}$ can be used to constrain carbon exchange above the canopy. The analysis is supported by the derivation of a new theoretical relationship that connects $ER_{atmos}$ and $ER_{forest}$ and by the use of a



mixed-layer model that reproduces the $O_2$ and $CO_2$ diurnal cycles coupled to the dynamics of the atmospheric boundary layer.
By combining the model with observations in a boreal forest during two contrasting summers of 2018 and 2019, we found three regimes during the day for $ER_{atmos}$.

We find that the entrainment of air from the free troposphere leads to a diurnal cycle in $ER_{atmos}$, resulting in three distinctive regimes: P1 at the start of the day, when the boundary layer has not yet started to grow, P2 when entrainment of air from the free troposphere into the boundary layer is dominant, and P3 at the end of the afternoon when entrainment becomes negligible. $ER_{atmos}$ can exhibit high values during P2 that cannot be attributed to an ER signal from a single process. During P3, $ER_{atmos}$ becomes closer to $ER_{forest}$, and is therefore more representative for the forest exchange.

The large diurnal variability in $ER_{atmos}$ shows that single height $O_2$ and $CO_2$ measurements are insufficient to be used as an indication for the $O_2/CO_2$ ratios of forest exchange. Our theoretical relationship between $ER_{atmos}$ and $ER_{forest}$ and model results show that the large diurnal variability is a result of the different behaviour of the $O_2$ and $CO_2$ diurnal cycle, which results in $ER_{atmos}$ values that cannot be attributed to a single process. To estimate the ER signal of the surface fluxes from above canopy measurements, $ER_{forest}$ should be used and therefore $O_2$ and $CO_2$ signals need to be measured at at least two heights, to allow fluxes to be calculated from the vertical gradient. A single measurement height of $O_2$ and $CO_2$ could still be used to indicate the presence of advection of other carbon sources. However, the resulting $ER_{atmos}$ signal should be analysed with care, by taking into account the diurnal variability and the fact that the resulting ER is not necessarily the average of the individually ER signals of the contributing processes.

When $O_2$ and $CO_2$ measurements are available from 2 heights, the relationship between $ER_{atmos}$ and $ER_{forest}$ during P2 and P3 could provide valuable information about the changes in large-scale carbon processes (e.g. entrainment) and their influence on the smaller scale processes of the surface. A discrepancy between $ER_{atmos}$ and $ER_{forest}$ shows that large scale processes occur together with small scale processes at the surface. The difference between $ER_{atmos}$ and $ER_{forest}$ should be analysed with care as the size of the difference is not a direct indication of the size of the influence of the large scale processes. Differences between $ER_{forest}$ and $ER_{atmos}$ could be caused by several factors: changes in the size of the entrainment flux, the net surface flux or the difference between the free troposphere and the boundary layer (the 'jump') for $O_2$ and/or $CO_2$, or changes in the jump ratio between $O_2$ and $CO_2$.

In conclusion, single height $O_2$ and $CO_2$ measurements need to be analyzed with care, accounting for their dependence on canopy processes (represented by $ER_{forest}$), but also for their capacity to integrate large scale processes resulting in values that cannot be attributed to a single process. To represent the forest exchange, the $ER_{forest}$ signal based on measurements at at least two heights should be used instead.





*Code availability.* The data used in this study are available from https://doi.org/10.18160/SJ3J-PD38 (Faassen and Luijkx, 2022). The model code for the CLASS model can be found in https://classmodel.github.io/

## Appendix A: Appendix

### A1 Evaluation of the theoretical relationship between $ER_{atmos}$ and $ER_{forest}$

In this Section, we analyze Equation 8 to explore the response of $ER_{atmos}$ to changes in the variables in this equation and to investigate when $ER_{atmos}$ aligns with $ER_{forest}$ and thereby accurately reflects local processes. Based on Equation 8, the $ER_{atmos}$ signal equals $ER_{forest}$ when the $\beta$ values of $O_2$ and $CO_2$ are equal. We can define four different regimes where the $\beta$ values change significantly. As depicted in Figure 1 we can define two regimes based on the entrainment velocity: an entrainment driven (left panels in Figure A1) and a photosynthesis driven regime right panels in Figure A1). To complete the analysis we considered two distinct cases for the jump of $O_2$ (top versus bottom panels).

Based on Equation 8 we systematically varied $\Delta_{(ft-bl)}CO_2$ and $(F_{CO_2})_s$ over plausible ranges and kept the other variables constant. As a result we derived $ER_{forest}$:$ER_{atmos}$ ratios for these four regimes, where a value of 1.0 now indicates that $ER_{atmos}$ is equal to $ER_{forest}$. The selected values and ranges for the four different cases were informed by initial conditions from the Hyytiälä case, studied in Faassen et al. (2023) and the corresponding model simulations presented in Section 3.2.4.

There are a few situations where the $\beta$ values of $O_2$ and $CO_2$ are equal and these are indicated in Figure A1 as the area between the black solid lines ($ER_{atmos}$ deviates <1% from $ER_{forest}$) and dashed lines ($ER_{atmos}$ deviates <10% from $ER_{forest}$):

1. During the photosynthesis dominant regime. When the entrainment velocity ($w_e$) is close to zero, both $\beta$ values become zero. This is likely at the end of the day (right panels in Figure 1).

2. When the $\beta$ values for $O_2$ and $CO_2$ become equal which happens when $\Delta_{(ft-bl)}O_2/\Delta_{(ft-bl)}CO_2 = ER_{forest}$. A specific case is when the $\Delta_{(ft-bl)}O_2 = \Delta_{(ft-bl)}CO_2$. In that case, the $ER_{forest}$ has to be 1.0 for the $\beta$ values of $O_2$ and $CO_2$ to become equal. The $\beta$ values of $O_2$ and $CO_2$ are become closer during the lower $O_2$ jump case (lower panels in Figure 1).

The last situation only occurs under very specific conditions when the ratio of the $O_2$ and $CO_2$ entrainment and surface fluxes are the same. This is visible in the left panels of Figure A1, where only a small part of the graph shows values of $ER_{atmos}$ close to $ER_{forest}$ (indicated by the area between solid lines). In contrast, during low entrainment velocities at the end of the afternoon, it is more likely that the $ER_{atmos}$ values become close to $ER_{forest}$, and this is shown by the larger area in the right panels of Figure A1. Low entrainment velocities could also occur when the growth of the boundary layer is reduces due to subsidence. During this study we will not focus on this specific case.

There are also differences between $ER_{atmos}$ and $ER_{forest}$ that arise from variations in the $\beta$ values. Figure A1 demonstrates that substantial differences between $ER_{atmos}$ and $ER_{forest}$ originate due to differences in the entrainment fluxes for both species.





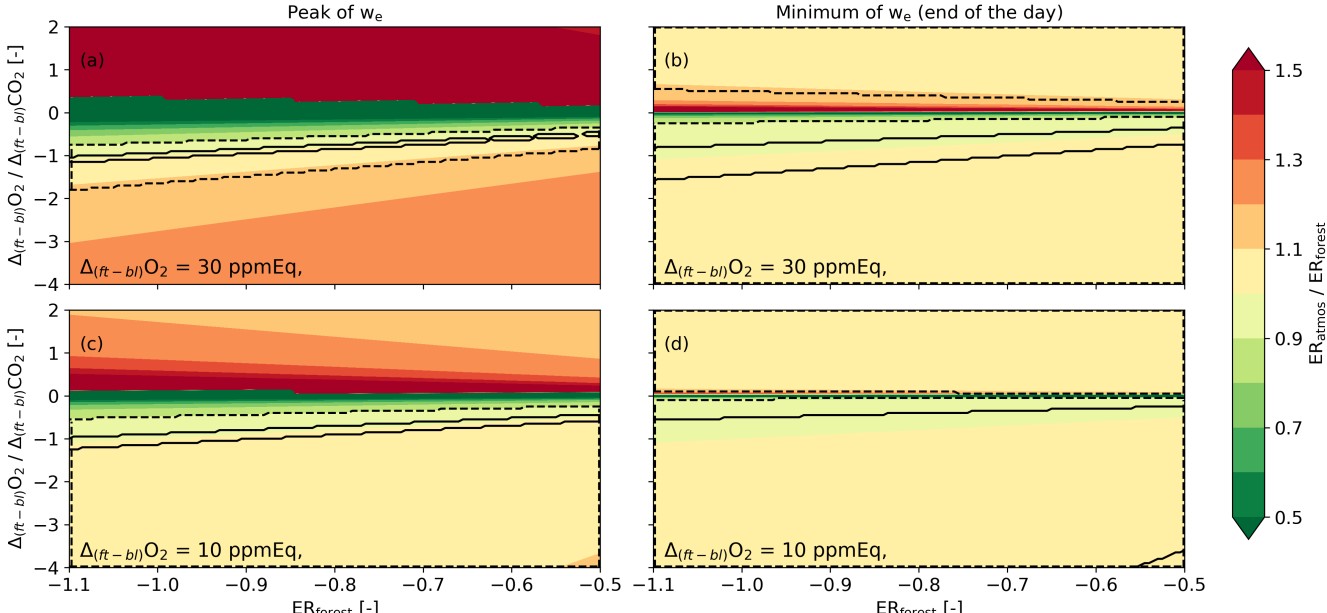

**Figure A1.** Analysis of Equation 8 for the entrainment- and photosynthesis-driven regimes. The ratio between $ER_{forest}$ and $ER_{atmos}$ is evaluated based on changes in $ER_{forest}$ and the ratio of the jumps of $O_2$ and $CO_2$ between the free troposphere and the boundary layer ($\Delta_{(ft-bl)}$) for 4 cases: with a high entrainment velocity ($w_e = 0.10$ m s$^{-1}$) (left panels) and a low entrainment velocity ($(w_e = 0.01$ m s$^{-1}$) (right panels) and for situations with a high $O_2$ jump ($\Delta_{(ft-bl)}O_2 = 0.30$ ppmEq) (top panels) and a low $O_2$ jump ($\Delta_{(ft-bl)}O_2 = 0.10$ ppmEq) (bottom panels). The $O_2$ surface flux $F(O_2)_s$ is kept constant for all the panels, at 8.5 $\mu$mol m s$^{-1}$.

When $\Delta_{(ft-bl)}O_2$ exceeds $\Delta_{(ft-bl)}CO_2$, this implies a dominant entrainment flux of $O_2$ over $CO_2$ and $\beta_{O_2}$ deviates further from $\beta_{CO_2}$ (Equation 8). This effect is almost absent when the jumps themselves are lower, because the $ER_{atmos}$ / $ER_{forest}$ ratio stays around 1 (Figure A1c). Moreover, when $\Delta_{(ft-bl)}CO_2$ transitions from negative to positive, the sign of $\beta_{CO_2}$ also changes, subsequently elevating the $ER_{atmos}$ values (Equation 8).

$ER_{atmos}$ can also become smaller than $ER_{forest}$ when $\Delta_{(ft-bl)}CO_2$ is larger than $\Delta_{(ft-bl)}O_2$ (Figure A1). This difference results in a large value for $\beta_{CO_2}$ compared to $\beta_{O_2}$, causing the $ER_{forest}$ value to be multiplied by a factor less than 1 and leading to a lower $ER_{atmos}$ value than $ER_{forest}$ (equation 8). By assessing $ER_{atmos}$ and $ER_{forest}$ values, we can see whether $\Delta_{(ft-bl)}O_2$ exceeds $\Delta_{(ft-bl)}CO_2$ ($ER_{atmos} > ER_{forest}$) or vice versa ($ER_{atmos} < ER_{forest}$).

This illustrative analysis, based on prescribed values in Equation 8 and Figure A1, provides an initial estimate of the variability in $ER_{atmos}$. However, it lacks insights into the diurnal behavior of the individual components of equation 8 and their potential combinations.



## A2 Implementation of $O_2$ in CLASS

The following equation shows the implementation of the tendency (change over time) of $O_2$ into CLASS:

$$\frac{dO_2}{dt} = \frac{F_{O_2(s)} - F_{O_2(e)}}{h} + adv_{O_2} \tag{A1}$$

where $F_{O_2(s)}$ is the net surface $O_2$ flux at the canopy, $F_{O_2(e)}$ is the $O_2$ entrainment flux, h is the boundary layer height and $adv_{O_2}$ is the advection term. The surface flux is calculated with equation 10 and the entrainment flux is based on the following equation (see also equation 2):

$$F_{O_2(e)} = -w_e \cdot \Delta_{(ft-bl)}O_2 \tag{A2}$$

where $w_e$ is the entrainment velocity and $\Delta_{(ft-bl)}O_2$ is the jump of $O_2$. The jump of $O_2$ was determined the same way as for $CO_2$, by tuning the jump until the decrease/increase in $CO_2/O_2$ matched during the entrainment dominant period.

## A3 Sensitivity analyses

### A3.1 Table with initialisation for the first two sensitivity analyses

**Table A1.** The initial conditions used for the three sensitivity analyses, compared to the initial conditions for the 2019 base case. The subscript (0) indicates the first time step.

| Variable | 2019 base | background air | | climate | |
| --- | --- | --- | --- | --- | --- |
| | | lower $\Delta_{(ft-bl)}$ ratio | lower intial $\Delta_{(ft-bl)}$ | high SMI | low SMI |
| $\Delta_{(ft-bl)}O_{2(0)}$ [ppmEq] | 30 | 30 | 10 | 2019 case | 2019 case |
| $\Delta_{(ft-bl)}CO_{2(0)}$ [ppm] | -8 | -20 | -5 | 2019 case | 2019 case |
| $\theta_0$ [K] | 285.2 | 2019 case | 2019 case | 290 | 290 |
| Soil moisture [$m^3\ m^{-3}$] | 0.18 | 2019 case | 2019 case | 0.22 | 0.14 |

## A4 Validation of CLASS

Figures A3 and A2 present a comparison between the model output of CLASS and the corresponding measurements for the representative days of 2018 and 2019, assessing various parameters. Both figures demonstrate that the model compares well to the observed data. CLASS accurately follows the observed temperature increase (Figure A2a). A constant difference of 655 approximately 8K between 2018 and 2019 is seen for both model and observations. This persistent difference is attributed to a heat wave rather than a drought in Hyytiälä, as a drought would have intensified the divergence between the 2018 and 2019 simulations throughout the day. Moreover, CLASS adequately models specific humidity for both years, assuming an initial relative humidity of 80% for 2018 (Figure A2b). The sensible heat flux (Figure A2c) and latent heat (Figure A2d) exhibit



minimal differences between the 2018 and 2019 simulations. The accurate representation of atmospheric properties in CLASS
consequently results in a satisfactory comparison of the boundary layer height development for both years in comparison to
the observed data from radiosondes (Figure A3a)

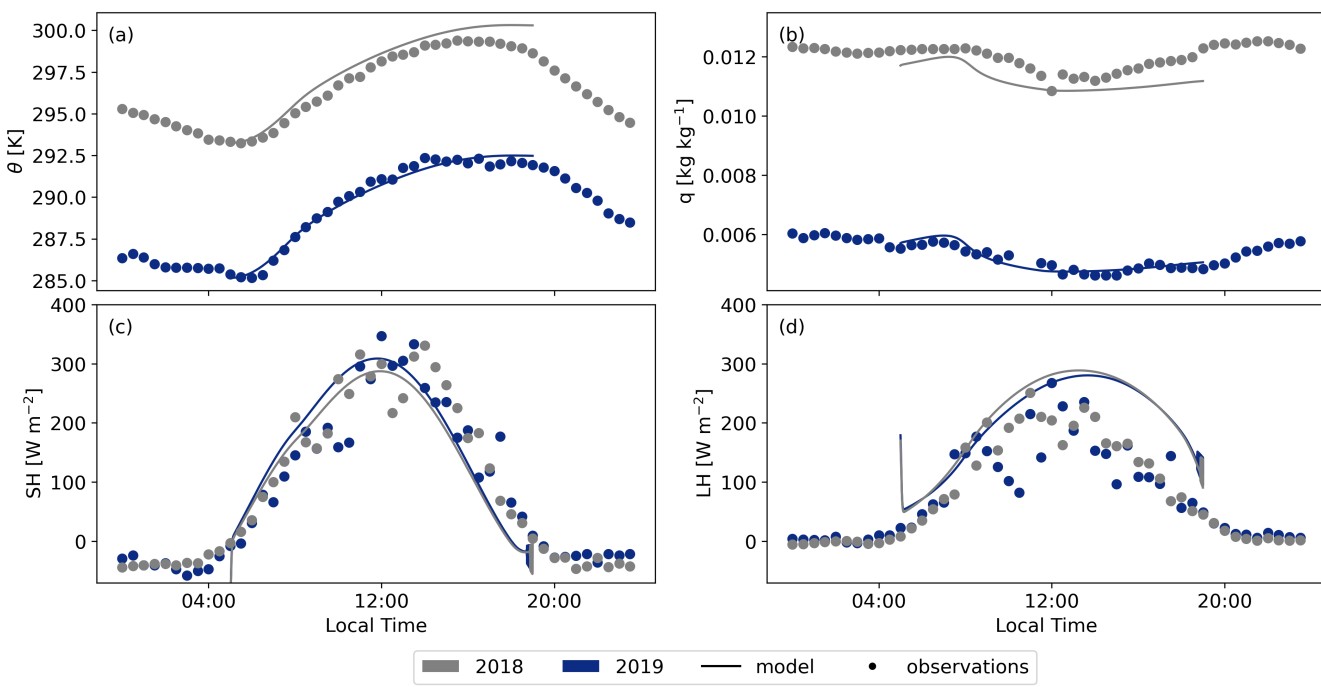

**Figure A2.** Comparison between the 2019 and 2018 case modelled with CLASS with the observational data for the potential temperature ($\theta$)
(a), specific humidity (q) (b), Sensible heat flux (SH) (c), Latent heat flux (LH) (d).

The various $CO_2$ fluxes simulated by CLASS exhibit a high level of agreement with the observational data for both 2018
and 2019 (Figure A3d and A3e). While there are subtle differences evident between the observations for the two years, CLASS
adeptly captures these nuances. Consequently, the model provides an accurate representation of plant behavior under both
normal and warmer conditions. The elevated temperatures (+8K) and slightly reduced soil moisture (-0.03 $m^3$ $m^{-3}$) con-
tribute to a slightly higher GPP and TER flux. Our study reaffirms that the vegetation in Hyytiälä did not undergo any stress
during the 2018 European drought, which would have resulted in a lower GPP and lower latent heat flux (Lindroth et al., 2020).

For the 2018 case, we only altered a few initial conditions (see Table C2). However, both the decrease in $CO_2$ and the in-
crease in $O_2$ during the day exhibit close similarity between the model and the observations. This outcome underscores that
even with minimal changes in the initial conditions for the 2018 case and keeping the other variables constant (e.g., the jumps),





we can successfully replicate a realistic new day based on the base case.

It is important to note that only the Net Ecosystem Exchange (NEE) data are obtained directly from Eddy Covariance measurements. The Gross Primary Production (GPP) is inferred from a light and temperature based function and the total ecosystem respiration is calculated as the residual between NEE and GPP (Kulmala et al., 2019; Kohonen et al., 2022). This distinction may explain the challenge in aligning the TER flux of the observations with the model, as the model exhibits notable discrepancies from the observations for both the 2018 and 2019 cases. The model's simulated respiration increase based on

temperature appears more extreme compared to the observations. However, several studies (Lindroth et al., 2008; Gao et al., 2017; Heiskanen et al., 2023) indicate that the model's increase in TER between 2018 and 2019 is slightly too high, while the change based on observations is too low. As a result, it is plausible that the true respiration flux lies somewhere between the model output and the observational data.

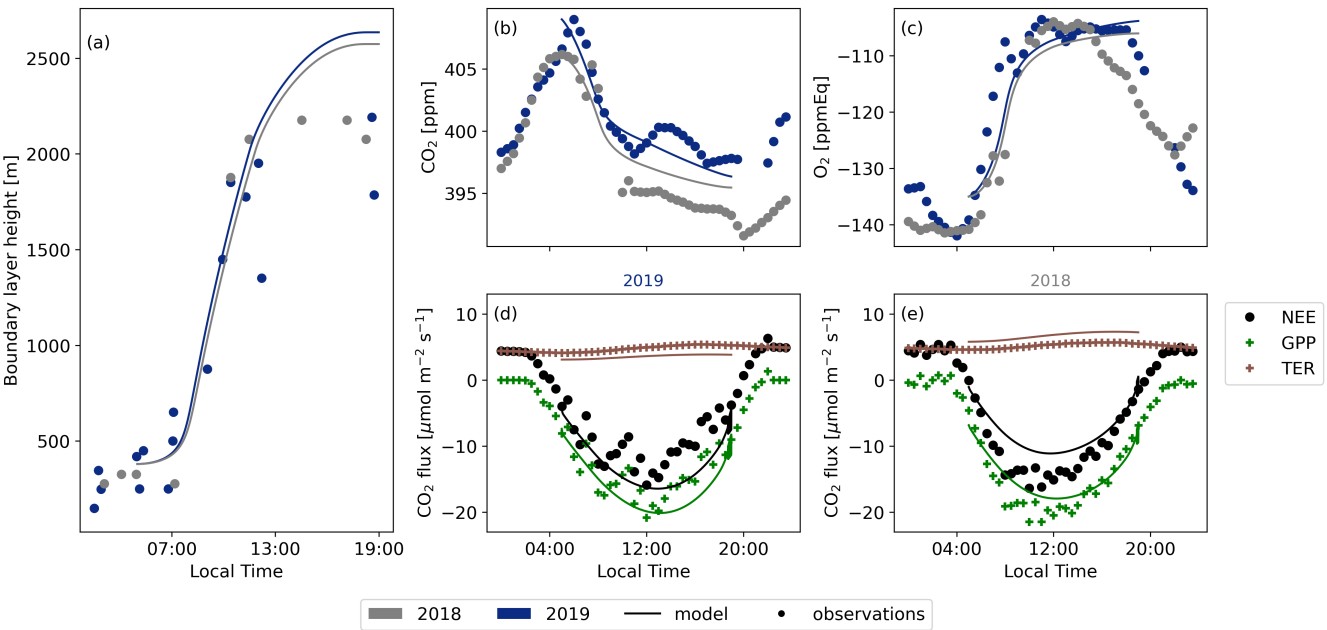

**Figure A3.** Comparison between the 2019 and 2018 case modelled with CLASS with the observational data for the boundary layer height (a), $CO_2$ (b), $O_2$ (c), the 2019 $CO_2$ surface fluxes (d) and the 2018 $CO_2$ surface fluxes (e).

**Appendix B: Figures**

**Appendix C: Tables**



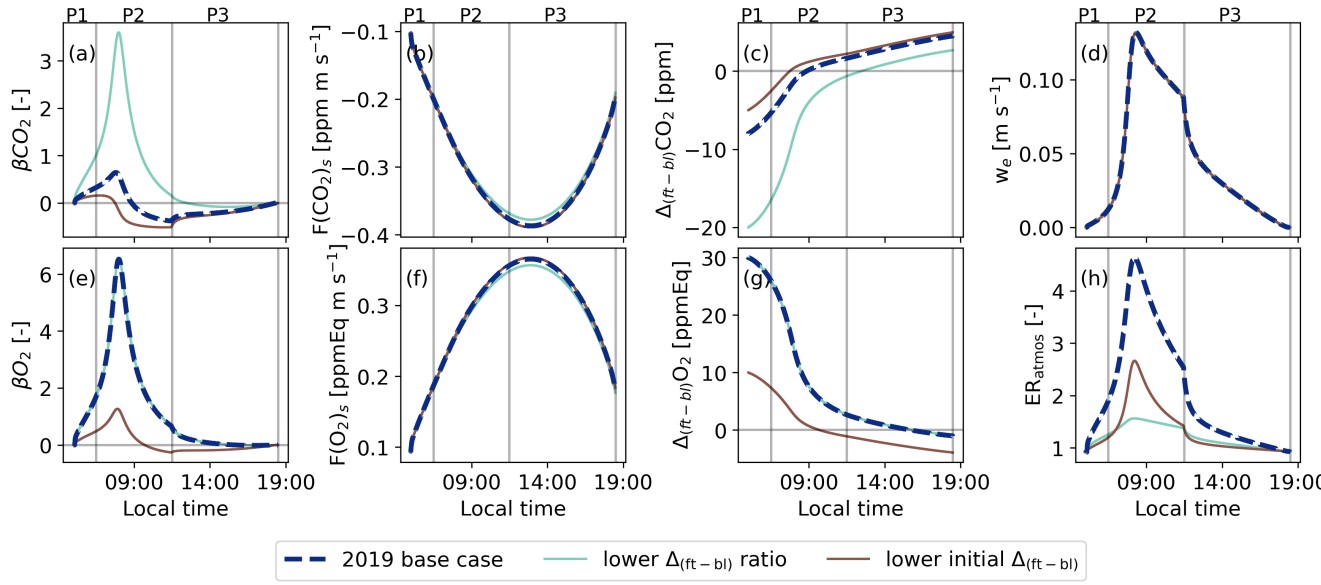

**Figure B1.** Similar to Figure 5 but now for the base case (2019) and the background sensitivity studies with a lower jump ratio between $O_2$ and $CO_2$ (lower $\Delta_{(ft-bl)}$) and with a lower initial jump for $CO_2$ (lower initial $\Delta_{(ft-bl)}$). The diurnal variability of the exchange ratio of the atmosphere is now added ($ER_{atmos}$: (h))

Table C1: Initialisation of the CLASS model for the base case of 2019, based on 10-07-2019. The initialisation is based on the SMEAR II data (Hari et al., 2013), our OXHYYGEN campaign data (radiosondes or $O_2$ and $CO_2$ measurements) (Faassen et al., 2023) and studies that show ranges for parameters for the plants and soil (Lindroth et al., 2008; ECMWF IV, 2014; Vilà-Guerau de Arellano et al., 2015) .

| Parameter [Source] | Description | Initial value |
|---|---|---|
| Lat | Latitude [deg] | 61.51 |
| Lon | Longitude [deg] | 24.17 |
| DOY | Day of year [-] | 191 |
| $t_0$ | Starting time [UTC] | 3 |
| $h_1$ | Initial boundary layer height [m] | 380 |
| $h_2$ | Height of the residual layer [m] | 2016 |
| P | Surface pressure [hPa] | 988.72 |
| *Temperature:* | | |
| $\theta_0$ | Initial potential temperature [K] | 285.15 |
| $\Delta\theta_0$ | Initial potential temperature jump [K] | 2.4 |

*Continues next page*



| | | |
|---|---|---|
| $\lambda\theta_1$ | Potential temperature lapse rate of residual layer [K m$^{-1}$] | 0.0023 |
| $\lambda\theta_2$ | Potential temperature lapse rate of free troposphere [K m$^{-1}$] | 0.0057 |
| *Specific humidity:* | | |
| $q_0$ | Initial specific humidity [kg kg$^{-1}$] | 5.7 x 10$^{-3}$ |
| $\Delta q_0$ | Initial specific humidity jump [kg kg$^{-1}$] | -1.2 x 10$^{-3}$ |
| $\lambda q_1$ | Specific humidity lapse rate of residual layer [kg kg$^{-1}$ m$^{-1}$] | -8.3 x 10$^{-7}$ |
| $\lambda q_2$ | Specific humidity lapse rate of free troposphere [kg kg$^{-1}$ m$^{-1}$] | -2.3 x 10$^{-6}$ |
| *Carbon:* | | |
| $CO_{2,0}$ | Initial $CO_2$ mole fraction [ppm] | 409 |
| $\Delta CO_{2,0}$ | Initial $CO_2$ jump [ppm] | -8 |
| $\lambda CO_2$ | $CO_2$ lapse rate of free troposphere [ppm m$^{-1}$] | 0 |
| *Oxygen:* | | |
| $O_{2,0}$ | Initial $O_2$ [ppm] | -135 |
| $\Delta O_{2,0}$ | Initial $O_2$ jump [ppm] | 30 |
| $\lambda O_2$ | $O_2$ lapse rate of free troposphere [ppm m$^{-1}$] | 0 |
| *Vegetation:* | | |
| LAI | Leaf Area Index [-] | 3.3 |
| $C_{veg}$ | Vegetation cover [-] | 0.9 |
| $r_{c,min}$ | Minimum resistance transpiration [s m$^{-1}$] | 500 |
| $r_{s,soil,min}$ | Minimum resistance soil evaporation [s m$^{-1}$] | 250 |
| $g_D$ | VPD correction factor for surface resistance [-] | 0.03 |
| $z_{0,m}$ | Roughness length for momentum [m] | 2.0 |
| $z_{0,h}$ | Roughness length for heat and moisture [m] | 2.0 |
| $\alpha$ | albedo [-] | 0.10 |
| $R_{10}$ | Respiration at 10 degrees [mg $CO_2$ m$^{-2}$ s$^{-1}$] | 0.148 |
| $g_m$ | Mesophyl conducatance [mm s$^{-1}$] | 2 |
| $T_{2gm}$ | reference temperature to calculate gm [K] | 305 |
| $C_\beta$ | Curvature of response curve to drought [-] | 0.15 |
| *Soil:* | | |
| $T_s$ | Initial surface temperature [K] | 287.7 |
| $T_{soil,1}$ | Initial top soil temperature [K] | 284.2 |
| $T_{soil,2}$ | Initial deeper soil temperature [K] | 282.0 |
| $w_{sat}$ | Saturated volumetric water content [m$^3$ m$^{-3}$] | 0.5 |

*Continues next page*



| $w_{fc}$ | Volumetric water content field capacity [$m^3$ $m^{-3}$] | 0.30 |
|---|---|---|
| $w_{wilt}$ | Volumetric water content wilting point [$m^3$ $m^{-3}$] | 0.08 |
| $w_g$ | Volumetric water content of top soil layer [$m^3$ $m^{-3}$] | 0.18 |
| $w_2$ | Volumetric water content of deeper soil layer [$m^3$ $m^{-3}$] | 0.12 |
| a | Clapp and Hornberger retention curve parameter [-] | 0.387 |
| b | Clapp and Hornberger retention curve parameter [-] | 4.05 |
| p | Clapp and Hornberger retention curve parameter [-] | 4 |
| $CG_{sat}$ | Saturated soil conductivity for heat [K $m^{-2}$ $J^{-1}$] | 3.22 x $10^{-6}$ |
| $C1_{sat}$ | Coefficient force term moisture [-] | 0.082 |
| $C2_{ref}$ | Coefficient restore term moisture [-] | 3.9 |
| $\Lambda$ | Thermal diffusivity skin layer [-] | 5 |

**Table C2.** Adjustments for the 2018 case (warm case) compared to the 2019 values shown in in table C1. Only the initial potential temperature ($\theta_0$), initial soil moisture ($w_g$) and $CO_2$ mole fraction ($CO_{2,0}$) are adjusted based on the aggregate of 28-07-2018 and 29-07-2018. It was assumed that the initial relative humidity stayed constant at 80% with increasing temperatures, therefore the initial specific humidity was also adjusted.

| Parameter | Description | Initial value |
|---|---|---|
| $\theta_0$ | Initial potential temperature [K] | 293.3 |
| $T_{soil,1}$ | Initial top soil temperature [K] | $\theta_0$ - 2 |
| $T_{soil,2}$ | Initial deeper soil temperature [K] | $\theta_0$ - 3 |
| $q_0$ | Initial specific humidity [kg $kg^{-1}$] | $f(\theta_0)$ |
| $w_g$ | Volumetric water content of top soil layer [$m^3$ $m^{-3}$] | $w_2$ - 0.04 |
| $w_2$ | Volumetric water content of deeper soil layer [$m^3$ $m^{-3}$] | 0.15 |
| $CO_{2,0}$ | Initial $CO_2$ mole fraction [ppm] | 406 |



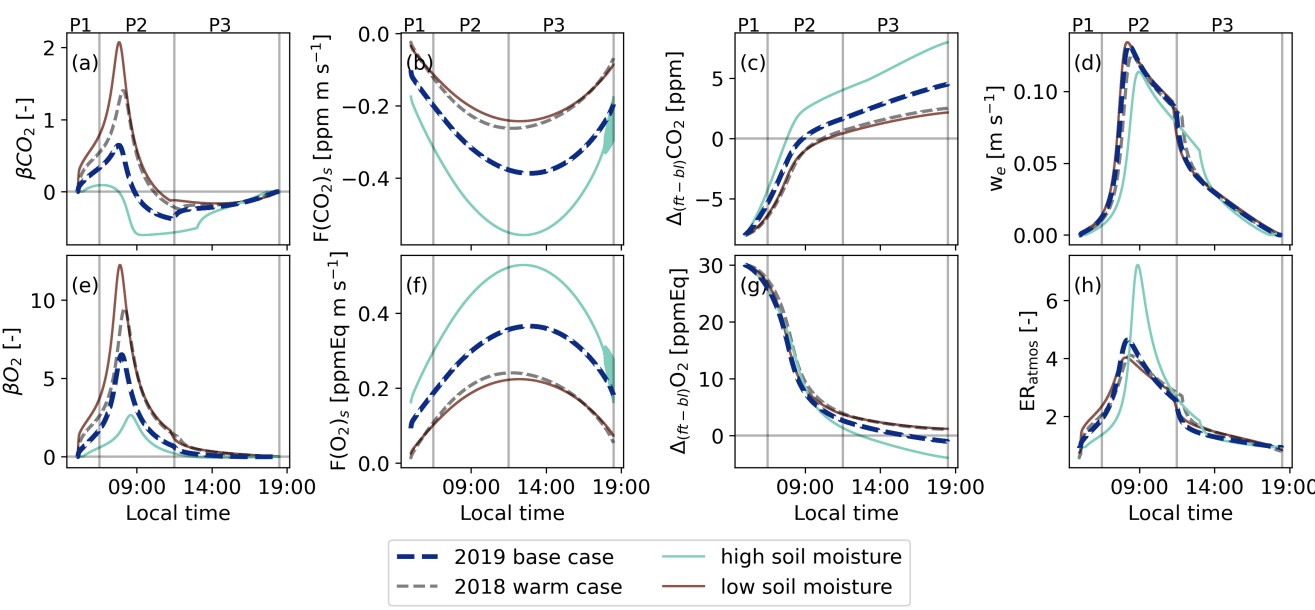

**Figure B2.** Similar to Figure 5 but now for the base case (2019) and the dry and warm sensitivity studies with a high soil moisture and a low soil moisture, both with higher air temperatures compared to the 2019 base case. The diurnal variability of the exchange ratio of the atmosphere is now added ($ER_{atmos}$: (h))



*Author contributions.* KAPF, JV, ITL and RG-A set up the model analysis. KAPF, ITL, JV and WP interpreted and discussed the methods and results. ITL designed the measurement campaign and conducted the $O_2$ and $CO_2$ measurements, and BGH conducted the radiosonde measurements with input from JV and support from IM. KAPF and ITL wrote the manuscript with input from all co-authors.

*Competing interests.* There are no competing interests.

*Acknowledgements.* The authors would like to thank the following persons for their help during the measurement campaigns at Hyytiälä in 2018 and 2019: Janne Levula (previously at Institute for Atmospheric and Earth System Research (INAR) / Physics, Faculty of Science, University of Helsinki, Helsinki, Finland), Timo Vesala (Institute for Atmospheric and Earth System Research (INAR) / Physics, Faculty of Science, University of Helsinki, Helsinki, Finland), Linh N.T. Nguyen (previously at University of Groningen, Centre for Isotope Re-
search, Energy and Sustainability Research Institute Groningen, Groningen, the Netherlands), Bert Kers (University of Groningen, Centre for Isotope Research, Energy and Sustainability Research Institute Groningen, Groningen, the Netherlands) and Brian Verhoeven (former student at Meteorology and Air Quality, Wageningen University, Wageningen, the Netherlands). We acknowledge Ruben van 't Loo (student Meteorology and Air Quality, Wageningen University, Wageningen, the Netherlands) for his contribution to the data analysis. We have used ChatGPT as a language editor to improve the readability of certain parts of the manuscript. This work was supported with funding from
the Netherlands Organisation for Scientific Research for ITL (016.Veni.171.095 and VI.Vidi.213.143). The measurements at Hyytiälä are supported by the University of Helsinki via ICOS-HY funding and the GreenFeedBack project from the EU Horizon Europe – Framework Programme for Research and Innovation (no. 101056921).



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
