# Peer review of "Separating above canopy CO2 and O2 measurements into their atmospheric and biospheric signatures"

_EGUsphere, 2023_

## Editor Comment (EC1)

In this manuscript, Faassen et al. tackle two questions:

1) How is it possible that the exchange ratio of atmospheric oxygen and carbon dioxide (inferred from the covariation of measurements of the abundance of these species at a single height) might be far higher than expected?
2) How does the covariation-based exchange ratio relate to the exchange ratio of the underlying forest, and is the latter really captured by a flux-based (two-height) set of measurements?

The answers are derived primarily from the CLASS model, characterizing the dynamics of the atmospheric mixed layer, along with measurements made at the Hyytiala research site in Finland.

Overall, this paper is a thorough and thoughtful investigation of a complicated phenomenon. It is a natural continuation of the work done by Faassen et al. (2023) and addresses many of the questions that were either left open, or were answered somewhat speculatively in that work. I feel the conclusions are well-supported and the paper is worthy of publication.

The manuscript is long and dense and includes appendices that are themselves substantive. I have the feeling that the authors might be able to improve the efficiency and clarity of their presentation, but this would require a wholesale re-write, and the current version is more than adequate (with some minor revisions suggested below). That said, I do find the abstract overly detailed – it would benefit from substantial shortening. In addition, the discussion could benefit from an effort to make the structure clearer to the reader.

My main scientific concerns/questions

In this work, ERatm is based on a single altitude well above the canopy. What if ERatm were based instead on a single altitude *within* the canopy (or even just above the canopy)? i.e. How would the conclusions of this paper change if ERatm were to be replaced with an ER based on the slope of an O2 vs CO2 plot made with data collected within the canopy? Perhaps a covariance-based ER is not intrinsically limited, but instead the limitations arise when the covariance data are collected far above the canopy. This might explain why the Hytialla data show ERatm values that are not observed at other locations.

My other questions center on the "jumps" between the atmospheric boundary layer and the free troposphere. These jumps are central to the explanation of the diurnal cycles in $ER_{atmos}$ (and by extension, the difference between $ER_{atmos}$ and $ER_{forest}$). The sensitivity analysis done is very valuable. My concern is that these jumps seem to be chosen *ad hoc*. I don't doubt that jumps exist, and I also am convinced by this work that they, along with entrainment, are the explanation for the observed values of $ER_{atmos}$, but the empirical evidence of jumps shown in Figure 2 is far from compelling. The data are consistent with the black-line conceptual models shown, but they are equally consistent with smaller (or non-existent) jumps and smoothly-varying lapse rates within the free troposphere. For example, the 10:25 LT trace from 2018 shows a "jump" at 1000m that's at least as big as some of the jumps depicted in the conceptual models. Likewise, the 9:56 LT trace from 2019 has a much bigger discontinuity at 1900m than it does at 1450m.

Furthermore, the authors show that the jump *ratios* depend quite strongly on the composition of the free troposphere background air.  Yes, if the background/free-troposphere air has very different O2 and CO2 values from what's in the boundary layer, then entrainment will result in extreme $ER_{atmos}$ values, but are the free-troposphere values explored in section 5.3 (and the sensitivity analyses) realistic?   At least the authors acknowledge the need for direct O2 and CO2 measurements in future campaigns (line 473) but perhaps this need should be emphasized.

Beyond these concerns, here are numerous minor editorial comments and suggestions:

In section 2, the authors refer to "mixed layer theory".  I my mind, this isn't a theory- it's a model. It's a particular representation of the mixed layer, choosing to include some processes and not others, based on educated judgement of what is important and what isn't.  If "mixed layer theory" is a widely used term (with which I'm simply not familiar), then stick with it, but otherwise, in the section title and throughout the paper, please replace "theory" with "model".

The first time "surface" is used, it should be explicitly defined.  Is it the surface of the soil?  The top of the canopy?  Likewise, on line 134, how is "above the canopy" defined?

L8: should read "measured at a single height"

L9: Should read "with the goal of relating the ERatmos signal to the ERforest signal and understanding the"

L20: should read "rarely represents ERforest directly and"

L23: should read "we recommend always measuring"

L26: should read "land use change emissions, moderated by uptake"

L27: comma after "oceans"

L28: should read "a valuable tracer, enhancing"

L29: remove the comma after "exchange"

L32: comma between "CO2" and "represents"

L33: should read "allow us to"

L44: should read "available instruments do not allow eddy covariance (EC)"

Fig1 caption: should read "over time can lead to"

L74: should read "that have not yet been measured"

L76: should read "In this study, we aim to"

L77: should read "and we propose a new"

L78: should read "measurements can be employed"

L79: should read "aforementioned limitations." And " whether the ERatmos signal constrains boundary layer dynamics, and we identify"

L89: should read "with the model CLASS (Sect. 3.)  We then show the model"

L91: should read "represents forest exchange (Sect. 4.) Next, we place"

L92: should read "(not) be used (Sect.5.)  Finally"

Eq. 1:  The "primed" terms should be explained/defined.

L109: should read "represent large scale"

L115: should read "associated normally with high pressure systems.  We assume wsub is negligible."

Eq.3:  Shouldn't there should be a subscript on the phi that is in time derivative?  We are left wondering if this is phi_bl or phi_ft?

Line 123: should read "layer height (dh/dt) effectively determines the entrainment velocity, and by extension the entrainment"

Eq. 4:  the last w should have a subscript consistent with eq. 2 (i.e. both should be either w_sub or w_s).  Also, in the denominator, it's not clear what the delta refers to.  The difference between where and where?

Line 127:  w_s or w_sub (like equations 2 & 4)?

Line 130:  As mentioned above, "Theoretical" versus "Modeled"

Line 143: Should read "According to the mixed-layer model described above"

Line 147: should read "term in Eq. 1 here, but we will add it later (Eq. 9).

Line 149/150: should read "definition of ERforest (Eq. 5) with Eq. 2 allows us to rewrite Eq. 7 as:"

Line 154: should read "and ERforest within the mixed layer model."

Line 156: should read "effect of other large scale processes such as advection of O2"

Eq. 9: Somehow, the authors should indicate that they're just introducing advection as an example of what one *could* do.  They can refer to section 5.2 in which they explicitly neglect advection. Whatever they decide, it should resolve the peculiarity of introducing terms which never get used.

Line 160: should read "values are of particular importance here: When the"

Line 163: The section that follows doesn't describe the CLASS model; it describes measurements.

Line 188: should read "et al., 1988).  We use conserved"

Line 188-190:  I simply don't understand this sentence.  How does the use of mole fractions imply that tracers are well-mixed?

Line 197: should read "time of 6 seconds, effectively averaging over 10m of altitude.

Line 213: should read "law of diffusion, based on the difference"

Line 217: should read "TER fluxes.  The differences between"

Line 237/238: should read "sizes and signs, each with their own ER"

Line 247: what is meant by "The final initial and boundary conditions"? Is "final initial" something in particular, or is it a list of three things (final conditions, initial conditions and boundary conditions)?

Line 259: should read "Combe et al., 2015), given that"

Line 265 and following: should read "and ERforest. Specifically, we looked at changes in ERatmos resulting from changing the different components of Eq. 8. The first sensitivity analysis uses the 2019 base case and investigates the effect of background air with a different composition by altering the initial jumps of O2 and CO2. By only changing"

Line 283: should read "While there is limited"

Line 295: I'm puzzled by "an increasing net CO2 flux out of the forest" since in panel d of Fig. 4, CO2 is dropping during P1.

Figure 4: In panel B there are two lines across the bottom. Presumably these are the values of ERforest, but they are unlabeled and the axis is ERatm (rather than a generic ER), so it's confusing. Please clarify.

Figure 5: The caption doesn't correctly describe what is in panel E. Please correct.

Line 229: should read "exhibit higher values than the model predictions of Sect. A1 because"

Line 345-346: should read "cases where ERatmos could equal ERforest if large-scale conditions were to change.

Line 347: should read "(Sect. 4.3.1), and changes in climate (soil moisture"

Line 356: should read "dominant and closer to"

Line 383: I'm pretty sure Figure 6 does not show energy balance closure.

Line 384: add a comma after "result"

Line 385: should read "the respiration, up to a threshold"

Line 386: should read "surface fluxes and an enhanced sensible heat flux. This will increase the boundary"

Line 392-393: should read "focus on two particular locations in the parameter space shown in Figure 7:"

Line 394: Begin a new paragraph with "A lower soil moisture..."

Line 395: should read "decreases ERatmos during P2 and increases ERatmos during P3"

Line 398: should read "and CO2 change more slowly and remain"

Line 401: add a comma after "result"

Line 402: should read "similar to the"

Line 426-427: I am not sure if you are saying ERa is set by nitrogen content in the leaf and light striking the leaf (i.e. "leaf level" applies to both nitrogen and light), or whether you are saying ERa is set by nitrogen content of the whole plant and light that strikes the leaf. Please clarify.

Line 430: should read "how ERatmos can change during"

Line 437: should read "ERforest. This also has the potential to improve estimates of the global biospheric ER, currently taken to be 1.1 (Severinghaus, 1995)"

Line 450: should read "2022a). However, caution should"

Line 452-453: should read "advected air. In addition to the surface and entrainment influences, ERatmos also depends on the magnitude of the advected flux. This is because mixing two ER"

Line 454: should read "of two sources with"

Line 455: should read "ERatmos values. A solution could be to include other tracers in the"

Line 459: remove the comma after "processes"

Line 460: should read "signal. During the day,"

Line 465: should read "ERforest may be due to either"

Line 466-467: should read "ratio. If the cause is the former (low BetaCO2), the ERatmos signal during P3 should be closer to ERforest. If the latter (a high jump ratio), ERatmos should remain well above ERforest in P3."

Line 473: should read "recommend that future measurement campaigns include"

Line 479: should read "In the absence of observational"

Line 485: What exactly is caused by mesoscale and synoptic processes? Subsidence, or the existence of the jump? Is it correct to say that they can cause subsidence that in turn creates a jump? Please clarify.

Figure 8: This is a valuable and information-rich figure, but the legend in Panel A is a bit confusing since there's no dark green in that panel. I suggest changing the green bar in the legend to light green (to match the light gray), and add something to the caption describing the difference between dark and light colors. Also, the arrows to/from the trees are a bit confusing since they only apply to a period of net respiration. I suggest taking out the green arrow and adding a few words to the caption description of Panel A saying it depicts only CO2 fluxes/abundances during a period of net respiration.

Line 492: By eye, ERforest seems to be less than 1.0 in all of the panels (the green arrows are always longer than the gray arrows.) Isn't this the case?

Line 495: should read "This can occur for example, when the"

Line 504-505: should read "found a non-linear relationship between O2 and other tracers that was difficult to explain. While"

Line 519: should read "constant ERatmos value."

Line 522: remove the comma after "that"

Line 526: should read "ERforest are likely rare."

Line 546: should read "Additional tracers can strengthen this approach. Del13C, "

Line 554: should read "ERatmos determined from the time dependence of O2 and CO2"

---

## Author Comment (AC1)

**RC1**: 'Comment on egusphere-2023-2833', Anonymous Referee #1, 28 Jan 2024

Thank you for the opportunity to review this manuscript. The authors present a thorough comparison between ERatmos and ERforest methodologies in quantifying the exchange of O2 and CO2 above a forest canopy. They demonstrate that ERatmos could be significantly influenced by entrainment, which results in unrealistic values. Consequently, the authors recommend against using ERatmos for constraining O2 and CO2 exchanges at a local scale, advocating instead for measurements at multiple heights to more accurately derive ERforest.

Entrainment significantly influences atmospheric composition within the boundary layer and is a well-researched phenomenon. However, this study stands out as the first, to my knowledge, that specifically addresses the impact of entrainment on O2 and CO2 exchanges. This represents a notable contribution to the field. This study suggests careful selection of O2 and CO2 measurements at single heights is required to correctly represent the biological exchange between O2 and CO2 in forest setting. This consideration is equally important in urban and other backgrounds, particularly for studies focusing on exchange ratios over smaller spatio-temporal scales. Given its importance and novelty, I recommend the acceptance of this study after the following issues are addressed.

We thank the reviewer for their assessment of our manuscript. We will address the remaining issues below.

Major comments:

1.   Is it possible for the effects of advection to be counterbalanced by those of entrainment?  The observed discrepancies between ERatmos and ERforest might stem from both entrainment and advection processes (Equation 8). In Appendix A1, the authors analyze the influence of the entrainment coefficient ($\beta$) on ERatmos signals and discuss instances where ERatmos aligns with ERforest. However, the role of advection remains unclear. Can we rely on measurements taken at a single height when advection's impact is potentially neutralized by entrainment? This interaction might explain why ERatmos and ERforest yield similar results.

We agree with the reviewer that advection of $CO_2$ and $O_2$ can influence ERatmos significantly, as we already briefly discuss in section 5.2. To show in more detail how advection can impact the ERatmos signal we did two extra analyses where we added advection to the 2019 CLASS case, in which the advected air either originates from (1) another forest or (2) from a fossil fuel source.

The amount that is advected of a certain scalar ($\varphi$) into the CLASS model can be determined with the following equation:

$$adv(\varphi) = \frac{d\varphi}{dx} \cdot u \qquad (1)$$

Where $d\varphi$ is the net horizontal gradient of either $CO_2$ or $O_2$ between the control volume solved by CLASS and a location outside the control volume (either the other forest or fossil fuel location), $dx$ is the distance between the location of the CLASS run and the other location and $u$ is the wind speed. Note that we take a net advection term, representative for the transport driven by the wind vectors $u$ and $v$ in the $x$ and $y$ direction respectively. For these two extra analyses we made in total 6 theoretical cases, where we assumed a distance of 50 km for $dx$ and 6 m s$^{-1}$ for $u$. We used different values for $d\varphi$ for the 6 cases, based on the difference between the 2019 CLASS run and another CLASS run (Table 1).

Table 1 The $d\varphi$ of Equation (1) that we have used to detemine the size of advection of $O_2$ and $CO_2$ for two extra analyses (either advecting air from another forest or from a fossil fuel source) and their three theoretical cases. The advection for the other forest cases changes over time and the $dO_2$ values therefore indicate the start value and the end value. The advection for the fossil fuel source stays from 12:30 with the value indicated in the table.

| Advection case | $dO_2$ [ppmEq] (t=0, t=11:30) | $dCO_2$ [ppm] |
|---|---|---|
| | (1) Other forest: | |
| Adv less entr $O_2$ | (0.0,-16.2) | 0 |
| Adv lower init $O_2$ | (-15, -15) | 0 |
| Adv both $O_2$ | (-15.0, -31.2) | 0 |
| | (2) Fossil fuel source: | |
| Same advection | -19.01 | 17.1 |
| Lower advection | -10.2 | 10.7 |
| Higher advection | -30.8 | 25.6 |

**(1) Advection of air from a forest location**

We did 3 simulations with CLASS including advection from another forest source. In these cases, the resulting $O_2$ mole fractions are lower than in the base case. We have implemented this in three ways: 1a) using a lower initial $O_2$ jump (adv less entr $O_2$), 1b) using a lower initial $O_2$ mole fraction (adv lower init $O_2$) and 1c) using both a lower jump and a lower initial $O_2$ mole fraction (adv both $O_2$).

Figure 1 shows that advection of less $O_2$ from a location with a forest, decreases the $O_2$ mole fraction compared to the 2019 base case (figure 1b) and as a result decreases the ERatmos value (figure 1c). We implemented the advection to start at the beginning of the run and to stop after 11:30.

In case advection from another forest would counterbalance the effect of entrainment as the reviewer suggests, the advected air needs to have less $O_2$ (negative $dO_2$) compared to the 2019 base case (Table 1). Advection of air with less

O$_2$ would reduce the steep increase of O$_2$ in P2 and therefore decrease ERatmos and bring it closer to ERforest.

We find that from the 3 cases, the "adv both O$_2$" case results in the lowest ERatmos values (1.68) and relatively comes the closest to ERforest (0.94), but still does not reach the same value. For reaching similar values of ERatmos compared to ERforest, the advection has to be of almost similar size as the fossil fuel cases (see below). We can therefore say that advection can only bring ERatmos closer to ERforest under very specific conditions. It is therefore highly unlikely that advection can bring ERatmos to a value that is representative for the ERforest signal.

[Figure]

*Figure 1 Results from advection from another forest for three theoretical cases (Table 1) and its impact on the CO$_2$ mole fraction (a), O$_2$ mole fraction (b) and the ERatmos (c). The advection starts when the run starts and ends at 11:30 Local Time. The numbers in figure (c) are the slopes of the linear regression lines for the period between 6:30 LT and 11:30 LT (P2 in manuscript).*

It is important to note that the advected air does not only have a forest surface exchange signature, but it also includes the effects of entrainment that occurred on the location where the air is advected from, and entrainment can therefore never be fully excluded.

**(2) Advection of air from a fossil fuel source**

For the case with advection from a fossil fuel source, we have advected air from a fossil fuel source that has an exchange ratio of -1.38, which is similar to the global average fossil fuel mix. We have simulated 3 cases, with either low, middle or high source strengths. We now advect both O$_2$ (dO$_2$) and CO$_2$ (dCO$_2$) (Table 1).

Figure 2 shows that advection can create an ERatmos signal that is not directly representative of the ER signal of the fossil fuel source. It is rather a mixture of the advected air (O2$_{advected}$), surface forest exchange (O2$_{forest}$) and atmospheric mixing (O2$_{entrainment}$)and it therefore requires cautious interpretation, as it cannot be

straightforwardly associated with the advected air due to non-linearity. This can be seen in Figure 2c, in that the linear regression lines do not reach the value of -1.38. The ERatmos value even becomes lower than 1 with our "low advection case".

[Figure]

*Figure 2 Results from advection from a fossil fuel source for three theoretical cases (Table 1) and its impact on the $CO_2$ mole fraction (a), $O_2$ mole fraction (b) and the ERatmos (c). The advection starts at 12:30 LT. The numbers in figure (c) are the slopes of the linear regression lines for the 3 cases.*

We have decided not to include these sensitivity analyses in the manuscripts, since the manuscript was already dense in content.

2. Does entrainment exert a more pronounced impact during typical days? This modelling study is generally based on the mixed layer theory. In studies by Ishidoya et al. (2013, 2015), their analysis did not specifically distinguish between measurements on 'typical days' and 'non-typical days', and derive similar ERatmos and ERforest values.

In the studies by Ishidoya et al. (2013, 2015), these authors do not analyze individual days but average several days into a composite day to analyze ERatmos and ERforest. By creating a composite (or representative) day, different scenarios of entrainment are included and as a result the extreme values of ERatmos could be averaged out. For example, the initial jump changes per day which results in different ERatmos values per day (Figure B1). By including different scenarios with different initial jumps, the high ERatmos values will be averaged out. We discuss this in section 5.4.

In our previous study (Faassen et al. 2023) we also made a composite day that was the average of seven days. Within these seven days typical and non-typical days were included. For example, two days that had a high amount of clouds during the day are part of our composite day. Even by including non-typical days, we showed that the ERatmos (-2.28) could still significantly deviate from ERforest.

After reading this work, I am fully convinced the impact of entrainment should be considered on 'typical day'. However, it remains uncertain how this applies to specific instances, such as heavily polluted urban days or during extraordinary events like COVID-19 lockdowns, where mixed layer theory may not always hold. It would be beneficial for readers to understand the frequency and significance of entrainment during these atypical periods.

Entrainment also occurs on non-typical days. Entrainment is mainly the result of the combination of the growth of the boundary layer and the difference (jump) of a scalar between the boundary layer and the free troposphere (Equation 2). This holds for both typical and non-typical days and for scenarios where mixed-layer theory may not fully apply. Entrainment occurs for each of these options, it is only easier to be study under weak synoptic and mesoscale conditions (typical days). However, when a frontal system passes, entrainment occurs, but its contribution is less.

The strength and the significance of entrainment depends on the day. A clear example of the importance of entrainment of $CO_2$ during a polluted day is given by the study of Casso-Torralba et al., (2008). They analyzed diurnal cycles of $CO_2$ measured above a grassland at the Cabauw station in the Netherlands. The study shows that during a (typical) day without advection, entrainment is dominating the $CO_2$ signal in the early morning. However, during a polluted day when advection of air with higher $CO_2$ levels is present (non-typical day), the advected air masses with high $CO_2$-concentrations balance the combined effect of entrained air or low $CO_2$ and the uptake of $CO_2$ by the surface. As a result, the $CO_2$ mole fractions hardly change during the day. During the polluted day, entrainment is still important, but it is not the dominant process during the morning anymore.

In Figure 2a above in this reply, we show a first analysis of how ERatmos could respond during 'non-typical' days with polluted advected air, and show that ERatmos should also be handled with care on polluted days.

When assessing single-height measurements on non-typical days, can we still depend on ERatmos for accurate representation? While modeling these atypical days using the CLASS model might be challenging, I suggest the authors discuss these considerations, possibly in Section 5.2, to provide a more comprehensive perspective.

For a new study focusing on measurements in a forest in the Netherlands, we are currently studying an example of a 'non-typical' day that includes subsidence. During this day we measured $O_2$ and $CO_2$ from flask samples (Figure 3a), as well as the jumps based on aircraft sampling (Figure 3b). During this non-typical day, the ERatmos comes closer to ERforest and this gives the illusion that ERatmos starts to represent the surface processes.

However, when analysing the different components of Equation 8, we find that the large jumps of $O_2$ and $CO_2$ create large entrainment fluxes (Figure 4). Together with low surface fluxes due to a heat wave during the campaign day, the beta values for both $O_2$ and $CO_2$ are high. Therefore, Eratmos is not representing ERforest, but ERatmos rather represents the ratio between the entrainment fluxes. ERatmos can therefore also not give a correct representation of ERforest during non-typical days.

[Figure]

*Figure 3 Measurements during our campaign in a forest the Netherlands on 18 May 2022 for ERatmos (a) and vertical profile measurements from aircraft samples, including their jumps based on the measured boundary layer height (b). The jumps in panel b are -38.0 ppm and -6.1 ppm for $CO_2$ for 10:00 LT and 14:30 LT respectively and the jumps for $O_2$ are 47.2 ppmEq and 10.6 ppmEq for 10:00 LT and 14:30 LT. The figure is based on a manuscript that is currently in preparation.*

[Figure]

*Figure 4 The different components of equation 8 (similar to Figure 5 in the manuscript) for the campaign day in the Netherlands that was modelled with CLASS. The figure is based on a manuscript that is currently in preparation.*

To clarify this point we have added a few sentences on the effect for non-typical days to section 5.2.

**References**

Casso-Torralba, P., Vilà-Guerau de Arellano, J., Bosveld, F., Soler, M. R., Vermeulen, A., Werner, C., and Moors, E.: Diurnal and vertical variability of the sensible heat and carbon dioxide budgets in the atmospheric surface layer, Journal of Geophysical Research: Atmospheres, 113, 2008.

Faassen, K. A., Nguyen, L. N., Broekema, E. R., Kers, B. A., Mammarella, I., Vesala, T., Pickers, P. A., Manning, A. C., Vilà-Guerau de Arellano, J., Meijer, H. A., et al.: Diurnal variability of atmospheric $O_2$, $CO_2$, and their exchange ratio above a boreal forest in southern Finland, Atmospheric Chemistry and Physics, 23, 851–876, 2023.

Ishidoya, S., Murayama, S., Takamura, C., Kondo, H., Saigusa, N., Goto, D., Morimoto, S., Aoki, N., Aoki, S., and Nakazawa, T.: $O_2$:$CO_2$ exchange ratios observed in a cool temperate deciduous forest ecosystem of central Japan , Tellus B: Chemical and Physical Meteorology,750 65, 21 120, https://doi.org/10.3402/tellusb.v65i0.21120, 2013.

Ishidoya, S., Murayama, S., Kondo, H., Saigusa, N., Kishimoto-Mo, A. W., and Yamamoto, S.: Observation of $O_2$:$CO_2$ exchange ratio for net turbulent fluxes and its application to forest carbon cycles, Ecological Research, 30, 225–234, 2015

---

## Author Comment (AC2)

**RC2**: 'Comment on egusphere-2023-2833', Anonymous Referee #2, 10 Mar 2024

The $O_2/CO_2$ exchange ratio above a forest canopy is a valuable tracer for understanding carbon exchange at the air-sea and air-land interfaces. We conventionally used a constant for global application, but this ratio can change significantly on a regional scale, and the mechanisms are still unclear. This manuscript presents an insightful analysis of the dominant mechanisms that determine the $O_2/CO_2$ exchange ratio by either using observations from a single height or from multiple heights. This study highlights the complexity of only using $CO_2$ and $O_2$ measurements at a single height to quantify ecosystem carbon flux, pointing out the advantage of using measurements from multiple heights for a precise ratio estimate.

The manuscript provides a comprehensive study, and the line of thought is mostly clear. I believe this paper is of interest to the general audience of Biogeosciences. I only have several concerns regarding the model experiment designs and the readability of the paper. I recommend a minor revision before this paper can be considered for publication.

We thank the reviewer for their detailed assessment of our manuscript. We will address the remaining issues below.

Main concerns:

1. It appears to me that the CLASS model was run for only one day with a prescribed initial condition. These runs clearly do not reach a steady state. In this case, the result strongly stands on the initial condition (e.g., the initial jump).

The reviewer is indeed right that the diurnal cycle modelled by CLASS depends strongly on the used initial conditions. The study by Pino et al. (2012) also confirms this. Understanding the initial conditions is therefore crucial to understand the results of the model during the day. The high dependency on the initial conditions also means that a diurnal cycle does not reach a steady state.

I am curious about the rationale for the initial jumps used in this study.

The initial jumps are determined based on fitting the diurnal cycle of the model to the observations of the mole fractions of $O_2$ and $CO_2$, due to lack of direct observations of the jumps themselves for which one would need an aircraft or very tall tower. To make this clearer we added some extra information to line 249.

To make sure we do not fully depend our results on the initial jumps that are fitted to our data, we did the sensitivities analyses in which we changed the initial jumps (see section 4.3.1) and we discuss this in section 5.3.

I am also curious if the authors have tried to run the model for multiple days to reach a semi-steady-state and check if there are different results.

Our sensitivity analyses (section 4.3) are an illustration of how different initial conditions would influence the $O_2$ and $CO_2$ concentration, based on variations in the soil moisture and the jumps. These results therefore correspond to runs for multiple "days". The results of the sensitivity analyses are shown in the form of the budget analyses in figures B1 and B2 and in the form of the final ERatmos/ERforest ratios (figure 7). These sensitivity analyses show indeed a wide range of results, but the message remains the same: ERatmos only rarely directly represents ERforest.

To strengthen this argument, for a manuscript in preparation we are currently investigating $O_2$ and $CO_2$ measurements from a campaign that was done in the Netherlands in 2022. Measurements for both $O_2$ and $CO_2$ where made from flask samples taken below and above the boundary layer height (using aircraft). In our response to Reviewer #1, we show in figure 3 some first results from that new analysis. The jumps in the morning of this measurement campaign are 47.2 ppmEq and -38.0 ppm for $O_2$ and $CO_2$ respectively. Based on those measurements we have designed a numerical experiment with a new CLASS case (Figure 4 of our reply to Reviewer #1). The analysis shows similar results as our "lower jump ratio case" from Figure B1. This new campaign confirms that the sensitivity analyses from our study are a physically-sounded representation of the variability of ERatmos during days with different initial jumps of $O_2$ and $CO_2$.

To make it more clear that the sensitivity analyses can be seen as representative for other days we included some extra information in line 346.

2. I appreciate the detailed study on factors modulating the $ER_{atmos}$ and $ER_{forest}$. The result part, however, contains too many details and reads more like a technical report. I suggest improving the readability, by either revising the leading sentence of each section to sharply focus on the main result or adding a leading paragraph summarizing the main findings. The abstract section also contains too many technical details. I suggest shortening it significantly.

We agree with the reviewer that the manuscript is relatively technical, making it challenging to read. Since this type of study has not been done before for $O_2$, we have chosen to include quite some explanations and conceptual figures, which we hoped would facilitate the understanding of the material we present (e.g. section 2 on the fundamental concepts). We acknowledge that this might make it technical to read the manuscript. We have therefore initially aimed to not include too many detailed figures in the main text and have rather shown the elaborated details only in the supplement. Based on this comment, we have tried to improve the readability further by removing technical details from the abstract and by adding some explanatory notes in the beginning of certain sections.

Minor comments:

1. Figure 1 is a little bit unclear. Could you label thick arrows with $F(O_2)$s and $F(CO_2)$s. It is also not clear in the figure that $ER_{forest}$ is actually calculated from the gradient. The two-sided arrow across BL seems to suggest that O2 and CO2 entrainment are of similar magnitude.

We have adjusted Figure 1 following the comments from the reviewer. It is important to note that the size of the arrows of the fluxes in Figure 1 are not a representation of the real ratio between the $O_2$ and $CO_2$ fluxes. The sensitivity analyses clearly show that the ratio between the entrainment fluxes can change per day. We did not want to make Figure 1 only a representation of the 2019 case that was modelled with CLASS, but rather represent a typical diurnal cycle.

2. L61: Expand on 'small scale process' upon its first mention for clarity.

Small scale processes are first mentioned in line 45 and are specified as forest exchange that occurs in and below the canopy. With large scale processes we mean mainly the influence of entrainment and this was already clarified in line 58.

3. L74: Please elaborate on extreme conditions (i.e., low SMI, etc.)

We have added some elaboration on what we mean with extreme conditions to line 74.

4. L132: Modify Eq. 5 to reflect that $ER_{forest}$ is derived from a gradient.

We have added gradients to Equation 5, to make it clear that ERforest is derived from fluxes calculated from vertical gradients.

5. L139: It is not clear how you calculate DtO2 and DtCO2. Based on fit to high-resolution data? What's the time window?

The values for $DtO_2$ and $DtCO_2$ in Equation 6 are normally not directly determined, rather, ERatmos is the result from a linear regression of $O_2$ and $CO_2$ data over a certain time window (Ishidoya, 2013; Keeling and Manning, 2014). The slope of this linear regression represents ERatmos and therefore represents the change of $O_2$ over time (DtO2) compared to $CO_2$ (DtCO2). The fit of the linear regression can be applied to concurrent $O_2$ and $CO_2$ values when values for more than 1 time step are present. This does not necessarily need to be high resolution data.

The resolution of the data and the time window are important to understand which processes and how much detail is included in the resulting ERatmos value.

The high time resolution of the CLASS model (1 data point per 10 seconds) shows a more detailed diurnal cycle of ERatmos with higher values compared to the observational data (1 value per 30 minutes). The time window over which the linear regression is applied also changes the resulting ERatmos value (see the different periods in Table 1), because different processes become important during the day.

We added some clarification in Line 139 about the time resolution of our observations and our model output, as well as on the time window of the linear regression.

6. L336- 339: According to Fig. 4b, 2018 features a very low $ER_{forest}$ during the night. Could you comment on whether it is related to elevated soil temperatures that only matter at night?

As described in Lines 306-310, the diurnal cycle of ERforest quantifies the ratio between TER and GPP. In Line 336 we explain that the low ERforest in 2018 is caused by a higher air and soil temperature. In the early morning in the 2018 case, the high soil temperature increased the TER while the GPP stayed relatively low just after sun rise. An increased TER flux compared to the GPP flux results in a lower ERforest value (Figure 3).

To make this clearer, we have added some extra information to Line 337.

7. Figure 8: This figure needs extra details. Arrows indicate sunrise to sunset. Better labeled on the figure to make this point clear.

We agree with the reviewer that this figure could be improved. We have adjusted the legend, and added clarifications in the figure, including arrows that indicate sunset to sunrise. The darker colors represent the vertical profile just after sunset and the lighter color indicates the vertical profile at the end of the night, just before sunrise.

8. L527-528: The finding that $ER_{atmos}$ can be so large compared to $ER_{forest}$ stands on the assumption that there is no vertical gradient in $CO_2$ and $O_2$ within the BL. If the resolved $ER_{atmos}$ is based on using data that is very close to the top of the BL, this $ER_{atmos}$ can be more sensitive to entrainment, compared to other studies.

We agree with the reviewer that the vertical gradient of $CO_2$ and $O_2$ within the BL, specifically close to the surface, indeed affects ERatmos. Under convective conditions 70 to 80% of the boundary layer will be well-mixed and only a gradient will be present near the canopy and in the entrainment zone. Closer to the canopy the ERatmos value would be less influenced by entrainment compared to further away. In our previous study (Faassen et al. 2023) we discuss this and

show that the measurements at 125 m height gives an ERatmos value of 3.40, whereas the measurements at 23 m result in an ERatmos value of 2.28 for the same day. The ERatmos value closer to the canopy is lower and closer to ERforest. However, the ERatmos value of the 23 m level still is influenced by entrainment.

. We have added some information in Line 511 to make it clear that the height at which ERatmos is determined also influences the difference between ERatmos and ERforest.

9. Figure A2: Why the model results are not extended toward the beginning and the end of the day

The CLASS model is only valid when the sensible heat flux (SH) is larger than zero, because it uses the mixed-layer theory. The mixed layer theory assumes that the boundary layer is fully mixed, which happens during the day when the atmosphere is unstable. The SH needs to be larger than zero to produce buoyancy and therefore mixing. During the night, the SH is negative and therefore the mixed-layer theory does not hold.

To make clear that the model needs a SH larger than zero and therefore only runs during the day, we added some extra information to Line 436.

10. Figure A3: It seems like the model overestimates the BL height in the afternoon and around the sunset. The simulated $O_2$ concentration also seems to have a clear phase lag compared to observation. How would these affect the simulated ER?

We agree with the reviewer that the CLASS model indeed shows an overestimation of the boundary layer height in the afternoon and shows a delayed onset of the increase in the $O_2$ values compared to the observations (Figure A3). The overestimation of the boundary layer growth between 13:00 and 14:30 (Figure A3a) results in an overestimation of the entrainment velocity within this time frame (Equation 4). This will however hardly influence the $O_2$ or the $CO_2$ budget because the surface is already the most dominant process in this time frame (P3, see Figure 4d and 4e). The overestimation of the BL height is therefore not significantly influencing the ER signals.

The later onset of the increase in the $O_2$ mole fraction could indeed influence the ERatmos values, and the delayed increase in $O_2$ means that it does not completely match the timing of the change in $CO_2$ for the model, and therefore the ERatmos signal deviates compared to the observations. Table 1 also shows that there are differences between ERatmos values of the observations and the model, with the largest difference in P2 where the later onset has the most influence. However, in both cases we find that the ERatmos values can become

very large (above 2), even without a perfect match between the model and the observations.

11. The title reads like the paper is trying to falsify the idea of using single-height $CO_2$ and $O_2$ observations to constrain surface carbon exchange. However, this approach is still valid if it only uses nighttime data. Given the detailed model experiments conducted in this study, I suggest modifying the title to better represent the comprehensive details of this work.

We agree with the reviewer that the current title does not reflect all comprehensive results from the study. However, we did not want to use a title that is too detailed or technical, and we therefore avoided words like entrainment or ERatmos versus ERforest in the title. Considering the suggestions of the reviewer to reflect our work more comprehensively, we have changed the title to:

'Separating above canopy $CO_2$ and $O_2$ measurements into their atmospheric and biospheric signatures.'

We would also like to add that we are not sure if ERatmos measurements during the night could represent the surface processes and that a one measurement height approach is still valid during nighttime. Measurements above the canopy could decouple from the surface during nighttime or advection could influence the measurements.

**References:**

Faassen, K. A., Nguyen, L. N., Broekema, E. R., Kers, B. A., Mammarella, I., Vesala, T., Pickers, P. A., Manning, A. C., Vilà-Guerau de Arellano, J., Meijer, H. A., et al.: Diurnal variability of atmospheric O2, CO2, and their exchange ratio above a boreal forest in southern Finland, Atmospheric Chemistry and Physics, 23, 851–876, 2023.

Ishidoya, S., Murayama, S., Takamura, C., Kondo, H., Saigusa, N., Goto, D., Morimoto, S., Aoki, N., Aoki, S., and Nakazawa, T.: O2:CO2 exchange ratios observed in a cool temperate deciduous forest ecosystem of central Japan , Tellus B: Chemical and Physical Meteorology,750 65, 21 120, https://doi.org/10.3402/tellusb.v65i0.21120, 2013.

Keeling, R. F. and Manning, A. C.: Studies of Recent Changes in Atmospheric O2 Content, vol. 5, Elsevier Ltd., 2 edn., https://doi.org/10.1016/B978-0-08-095975-7.00420-4, 201

Pino, D., Vilà-Guerau de Arellano, J., Peters, W., Schröter, J., van Heerwaarden, C. C., and Krol, M. C.: A conceptual framework to quantify the influence of convective boundary layer development on carbon dioxide mixing ratios, Atmos. Chem. Phys., 12, 2969–2985, https://doi.org/10.5194/acp-12-2969-2012, 2012.

---

## Author Comment (AC3)

**EC1**: 'Comment on egusphere-2023-2833', Anonymous Referee #3, 25 March 2024

In this manuscript, Faassen et al. tackle two questions:
1) How is it possible that the exchange ratio of atmospheric oxygen and carbon dioxide (inferred from the covariation of measurements of the abundance of these species at a single height) might be far higher than expected?
2) How does the covariation-based exchange ratio relate to the exchange ratio of the underlying forest, and is the latter really captured by a flux-based (two-height) set of measurements?

The answers are derived primarily from the CLASS model, characterizing the dynamics of the atmospheric mixed layer, along with measurements made at the Hyytiala research site in Finland.

Overall, this paper is a thorough and thoughtful investigation of a complicated phenomenon. It is a natural continuation of the work done by Faassen et al. (2023) and addresses many of the questions that were either left open, or were answered somewhat speculatively in that work. I feel the conclusions are well-supported and the paper is worthy of publication.

The manuscript is long and dense and includes appendices that are themselves substantive. I have the feeling that the authors might be able to improve the efficiency and clarity of their presentation, but this would require a wholesale re-write, and the current version is more than adequate (with some minor revisions suggested below). That said, I do find the abstract overly detailed – it would benefit from substantial shortening. In addition, the discussion could benefit from an effort to make the structure clearer to the reader.

We thank the reviewer for their assessment of our manuscript. We acknowledge that it is indeed dense in content and fundamental concepts. We struggled especially to keep it short because we also wanted to introduce some theory that is not commonly used in the field of atmospheric mole fraction observations.

We have shortened the abstract and tried to remove technical details from it. Also, based on other reviewers' comments, we have introduced some lines in the beginning of the results sections to better structure it. Likewise, we have added a few lines with an overview of the structure in the beginning of the discussion section.

My main scientific concerns/questions
In this work, ERatm is based on a single altitude well above the canopy. What if ERatm were based instead on a single altitude within the canopy (or even just above the canopy)? i.e. How would the conclusions of this paper change if ERatm were to be replaced with an ER based on the slope of an O2 vs CO2 plot made with data collected within the canopy? Perhaps a covariance-based ER is not intrinsically limited, but instead the limitations arise when the covariance data are collected far above the canopy. This might explain why the Hytialla data show ERatm values that are not observed at other locations.

This is an interesting point. Unfortunately, it is not possible anymore to verify in-canopy values retrospectively for our campaigns in Hyytiala, that were done in 2018 and 2019. However, as we also wrote in response to point 8 of reviewer #2, we did compare the ERatmos values of the 125 m level with the 23 m level measurements. The canopy height at Hyytiala is around 18 meters. We note that the influence of entrainment on the measurements is smaller closer to the canopy at 23 m, but still present, and we get high values of 2.28 for ERatmos at that level. We have included this in line 511. The influence of entrainment on canopy level is for example also shown by the study of Patton et al. (2016) where they indicate that entrainment is also important at canopy level.

We would recommend to always use a height above the canopy to determine ERatmos, to make sure that the fluxes of the complete canopy are taken into account in the calculation. When measuring inside the canopy there is a risk that the fluxes above the measurement height are missed and the ER signal does not incorporate the complete GPP and TER fluxes. More research should be done on the implications of measuring the ERforest inside the canopy compared to just above the canopy.

My other questions center on the "jumps" between the atmospheric boundary layer and the free troposphere. These jumps are central to the explanation of the diurnal cycles in ERatmos (and by extension, the difference between ERatmos and ERforest). The sensitivity analysis done is very valuable. My concern is that these jumps seem to be chosen ad hoc. I don't doubt that jumps exist, and I also am convinced by this work that they, along with entrainment, are the explanation for the observed values of ERatmos, but the empirical evidence of jumps shown in Figure 2 is far from compelling. The data are consistent with the black-line conceptual models shown, but they are equally consistent with smaller (or non-existent) jumps and smoothly-varying lapse rates within the free troposphere. For example, the 10:25 LT trace from 2018 shows a "jump" at 1000m that's at least as big as some of the jumps depicted in the conceptual models. Likewise, the 9:56 LT trace from 2019 has a much bigger discontinuity at 1900m than it does at 1450m.

We acknowledge that we do not have observational evidence of the $O_2$ and $CO_2$ "jumps" which would have strongly improved our analysis. Note that in Figure 2, we only show observational evidence for the potential temperature profiles, which we could measure with radiosondes, but this is not possible for $O_2$ and $CO_2$ (even though we have actually tried for $CO_2$ in this campaign with very light weight $CO_2$ sensors, but they did not give accurate results). Since we don't have observed values for the $O_2$ and $CO_2$ jumps, and as the reviewer says, there can also be variable, we assess the influence of different values for the jumps in our sensitivity analyses in Section 4.3.1.

In a recent new campaign, we did manage to get observations for the jumps for $O_2$ and $CO_2$, based on flask samples collected from aircraft. This was in a forest in the Netherlands. In our response to reviewer #1, we have included preliminary results from that campaign, which confirm that the values that we use here are realistic.

The jumps of the potential temperature profile are based on the difference between the mixed layer and the lapse rate value at the boundary layer height. The boundary layer height is not chosen ad hoc, but determined with the parcel method (Kaimal and Finnigan,

1994). We added the reference to Kaimal and Finnigan, (1994) in Line 199. The radiosondes also do not represent the full extent of the boundary layer at the time specified in Figure 2, because they only go through one specific trajectory and therefore do not measure the complete mixed boundary layer above the forest. The vertical profile of potential temperature is therefore not perfectly representing the mixed layer.

Furthermore, the authors show that the jump ratios depend quite strongly on the composition of the free troposphere background air. Yes, if the background/free-troposphere air has very different O2 and CO2 values from what's in the boundary layer, then entrainment will result in extreme ERatmos values, but are the free-troposphere values explored in section 5.3 (and the sensitivity analyses) realistic? At least the authors acknowledge the need for direct O2 and $CO_2$ measurements in future campaigns (line 473) but perhaps this need should be emphasized.

As we stated above, indeed we think that our values are realistic, based on the new campaign. We have also introduced this around line 473 in the manuscript.

Beyond these concerns, here are numerous minor editorial comments and suggestions:

In section 2, the authors refer to "mixed layer theory". I my mind, this isn't a theory- it's a model. It's a particular representation of the mixed layer, choosing to include some processes and not others, based on educated judgement of what is important and what isn't. If "mixed layer theory" is a widely used term (with which I'm simply not familiar), then stick with it, but otherwise, in the section title and throughout the paper, please replace "theory" with "model".

Mixedlayer theory is the theory on which the mixed layer model is based. It is a commonly used term in boundary layer meteorology and we therefore keep it is as it is in the manuscript.

The first time "surface" is used, it should be explicitly defined. Is it the surface of the soil? The top of the canopy? Likewise, on line 134, how is "above the canopy" defined?

We have added the following line to the caption of Figure 1:
Note that the term "surface fluxes" refers to the fluxes from the surface layer, which includes the vegetation layer including the top of the canopy. The surface layer is the lowest 10% of the boundary layer where the surface directly influences the atmospheric boundary layer.

With "above the canopy" we mean that it is not inside that canopy. We explain this in line 134.

L8: should read "measured at a single height"

Changed accordingly.

L9: Should read "with the goal of relating the ERatmos signal to the ERforest signal and understanding the"

This sentence was changed while rewriting the abstract.

L20: should read "rarely represents ERforest directly and"

Changed accordingly (in a rewritten sentence).

L23: should read "we recommend always measuring"

Changed accordingly.

L26: should read "land use change emissions, moderated by uptake"

Changed accordingly.

L27: comma after "oceans"

Changed accordingly.

L28: should read "a valuable tracer, enhancing"

Changed accordingly.

L29: remove the comma after "exchange"

Changed accordingly.

L32: comma between "CO2" and "represents"

Changed accordingly.

L33: should read "allow us to"

Changed accordingly.

L44: should read "available instruments do not allow eddy covariance (EC)"

Changed accordingly.

Fig1 caption: should read "over time can lead to"

Changed accordingly.

L74: should read "that have not yet been measured"

We removed this part of the sentence, while adding some more details based on a comment from another reviewer.

L76: should read "In this study, we aim to"

Changed accordingly.

L77: should read "and we propose a new"

Changed accordingly.

L78: should read "measurements can be employed"

Changed accordingly.

L79: should read "aforementioned limitations." And " whether the ERatmos signal constrains boundary layer dynamics, and we identify"

Changed accordingly.

L89: should read "with the model CLASS (Sect. 3.) We then show the model"

Changed accordingly.

L91: should read "represents forest exchange (Sect. 4.) Next, we place"

Changed accordingly.

L92: should read "(not) be used (Sect.5.) Finally"

Changed accordingly.

Eq. 1: The "primed" terms should be explained/defined.

Changed accordingly.

L109: should read "represent large scale"

Changed accordingly.

L115: should read "associated normally with high pressure systems. We assume wsub is negligible."

Changed accordingly.

Eq.3: Shouldn't there should be a subscript on the phi that is in time derivative? We are left wondering if this is phi_bl or phi_ft?

This is the boundary layer value (well-mixed tracer), as defined with equation 1. We repeated it now here as well.

Line 123: should read "layer height (dh/dt) effectively determines the entrainment velocity, and by extension the entrainment"

Changed.

Eq. 4: the last w should have a subscript consistent with eq. 2 (i.e. both should be either w_sub or w_s). Also, in the denominator, it's not clear what the delta refers to. The difference between where and where?

Changed accordingly.

Line 127: w_s or w_sub (like equations 2 & 4)?

Changed accordingly.

Line 130: As mentioned above, "Theoretical" versus "Modeled"

This is theory, not modelled.

Line 143: Should read "According to the mixed-layer model described above"

Changed, but get theory, as this is not the model.

Line 147: should read "term in Eq. 1 here, but we will add it later (Eq. 9).

Changed accordingly.

Line 149/150: should read "definition of ERforest (Eq. 5) with Eq. 2 allows us to rewrite Eq. 7 as:"

Changed accordingly.

Line 154: should read "and ERforest within the mixed layer model."

Changed but kept theory.

Line 156: should read "effect of other large scale processes such as advection of O2"

Changed accordingly.

Eq. 9: Somehow, the authors should indicate that they're just introducing advection as an example of what one could do. They can refer to section 5.2 in which they explicitly neglect

advection. Whatever they decide, it should resolve the peculiarity of introducing terms which never get used.

Changed accordingly.

Line 160: should read "values are of particular importance here: When the"

Changed accordingly.

Line 163: The section that follows doesn't describe the CLASS model; it describes measurements.

Changed accordingly.

Line 188: should read "et al., 1988). We use conserved"

Changed accordingly.

Line 188-190: I simply don't understand this sentence. How does the use of mole fractions imply that tracers are well-mixed?

Agreed, we have removed this sentence since it was unclear and not necessary.

Line 197: should read "time of 6 seconds, effectively averaging over 10m of altitude.

Changed accordingly.

Line 213: should read "law of diffusion, based on the difference"

Changed accordingly.

Line 217: should read "TER fluxes. The differences between"

Changed accordingly.

Line 237/238: should read "sizes and signs, each with their own ER"

Changed accordingly.

Line 247: what is meant by "The final initial and boundary conditions"? Is "final initial" something in particular, or is it a list of three things (final conditions, initial conditions and boundary conditions)?

We have removed the word "final".

Line 259: should read "Combe et al., 2015), given that"

Changed accordingly.

Line 265 and following: should read "and ERforest. Specifically, we looked at changes in ERatmos resulting from changing the different components of Eq. 8. The first sensitivity analysis uses the 2019 base case and investigates the effect of background air with a different composition by altering the initial jumps of O2 and CO2. By only changing"

Changed accordingly.

Line 283: should read "While there is limited"

Changed accordingly.

Line 295: I'm puzzled by "an increasing net CO2 flux out of the forest" since in panel d of Fig. 4, $CO_2$ is dropping during P1.

We have changed "out of" to "into" since indeed the net flux of $CO_2$ is negative (uptake). See also Figure A3.
Figure 4: In panel B there are two lines across the bottom. Presumably these are the values of ERforest, but they are unlabeled and the axis is ERatm (rather than a generic ER), so it's confusing. Please clarify.

Good point, these lines are indeed ERforest. We improved Figure 4 by making the legends and axes more clear.

Figure 5: The caption doesn't correctly describe what is in panel E. Please correct.

Changed accordingly.

Line 229: should read "exhibit higher values than the model predictions of Sect. A1 because"

Changed accordingly.

Line 345-346: should read "cases where ERatmos could equal ERforest if large-scale conditions were to change.

This was rewritten.

Line 347: should read "(Sect. 4.3.1), and changes in climate (soil moisture"

This was rewritten.

Line 356: should read "dominant and closer to"

Changed accordingly.

Line 383: I'm pretty sure Figure 6 does not show energy balance closure.

Changed this.

Line 384: add a comma after "result"

Changed accordingly.

Line 385: should read "the respiration, up to a threshold"

Changed accordingly.

Line 386: should read "surface fluxes and an enhanced sensible heat flux. This will increase the boundary"

Changed accordingly.

Line 392-393: should read "focus on two particular locations in the parameter space shown in Figure 7:"

Changed accordingly.

Line 394: Begin a new paragraph with "A lower soil moisture…"

Changed accordingly.

Line 395: should read "decreases ERatmos during P2 and increases ERatmos during P3"

Changed accordingly.

Line 398: should read "and CO2 change more slowly and remain"

Changed accordingly.

Line 401: add a comma after "result"

Changed accordingly.

Line 402: should read "similar to the"

Changed accordingly.

Line 426-427: I am not sure if you are saying ERa is set by nitrogen content in the leaf and light striking the leaf (i.e. "leaf level" applies to both nitrogen and light), or whether you are saying ERa is set by nitrogen content of the whole plant and light that strikes the leaf. Please clarify.

Changed accordingly.

Line 430: should read "how ERatmos can change during"

Changed accordingly.

Line 437: should read "ERforest. This also has the potential to improve estimates of the global biospheric ER, currently taken to be 1.1 (Severinghaus, 1995)"

Changed accordingly.

Line 450: should read "2022a). However, caution should"

Changed accordingly.

Line 452-453: should read "advected air. In addition to the surface and entrainment influences, ERatmos also depends on the magnitude of the advected flux. This is because mixing two ER"

Changed accordingly.

Line 454: should read "of two sources with"

Changed accordingly.

Line 455: should read "ERatmos values. A solution could be to include other tracers in the"

Changed accordingly.

Line 459: remove the comma after "processes"

Changed accordingly.

Line 460: should read "signal. During the day,"

Changed accordingly.

Line 465: should read "ERforest may be due to either"

Changed accordingly.

Line 466-467: should read "ratio. If the cause is the former (low BetaCO2), the ERatmos signal during P3 should be closer to ERforest. If the latter (a high jump ratio), ERatmos should remain well above ERforest in P3."

Changed accordingly.

Line 473: should read "recommend that future measurement campaigns include"

Changed accordingly.

Line 479: should read "In the absence of observational"

Changed accordingly.

Line 485: What exactly is caused by mesoscale and synoptic processes? Subsidence, or the existence of the jump? Is it correct to say that they can cause subsidence that in turn creates a jump? Please clarify.

We meant the subsidence. Changed accordingly.

Figure 8: This is a valuable and information-rich figure, but the legend in Panel A is a bit confusing since there's no dark green in that panel. I suggest changing the green bar in the legend to light green (to match the light gray), and add something to the caption describing the difference between dark and light colors. Also, the arrows to/from the trees are a bit confusing since they only apply to a period of net respiration. I suggest taking out the green arrow and adding a few words to the caption description of Panel A saying it depicts only CO2 fluxes/abundances during a period of net respiration.

We have modified the figure according also to comments from reviewer #2. The flux arrows are indeed for net respiration, since it is during the night time. We have also updated the caption to clarify.

Line 492: By eye, ERforest seems to be less than 1.0 in all of the panels (the green arrows are always longer than the gray arrows.) Isn't this the case?

The reviewer is indeed correct that the green arrows (indicating $O_2$) are always longer than the grey arrows (indicating $CO_2$). However, this means that ERforest is larger than 1.0 because ERforest is $O_2/CO_2$. This decision was made because several studies have shown that ERr (ER of respiration, the ERforest during the night) is larger than 1.0 (Ishidoya et al., 2013; Angert et al., 2015; Hilman et al., 2022). This was briefly mentioned in line 504. We added the above-mentioned studies for extra clarity.

Line 495: should read "This can occur for example, when the"

Changed accordingly.

Line 504-505: should read "found a non-linear relationship between O2 and other tracers that was difficult to explain. While"

Changed accordingly.

Line 519: should read "constant ERatmos value."

Changed accordingly.

Line 522: remove the comma after "that"

Changed accordingly.

Line 526: should read "ERforest are likely rare."

Changed accordingly.

Line 546: should read "Additional tracers can strengthen this approach. Del13C, "

Changed accordingly.

Line 554: should read "ERatmos determined from the time dependence of O2 and CO2"

Changed accordingly.

**References:**

Angert, A., Yakir, D., Rodeghiero, M., Preisler, Y., Davidson, E. A., and Weiner, T.: Using O2 to study the relationships between soil CO2 efflux and soil respiration, Biogeosciences, 12, 2089–2099, https://doi.org/10.5194/bg-12-2089-2015, 2015.

Hilman, B., Weiner, T., Haran, T., Masiello, C. A., Gao, X., and Angert, A.: The apparent respiratory quotient of soils and tree stems and the processes that control it, Journal of Geophysical Research: Biogeosciences, 127, e2021JG006 676, 2022.

Ishidoya, S., Murayama, S., Takamura, C., Kondo, H., Saigusa, N., Goto, D., Morimoto, S., Aoki, N., Aoki, S., and Nakazawa, T.: O2:CO2 exchange ratios observed in a cool temperate deciduous forest ecosystem of central Japan , Tellus B: Chemical and Physical Meteorology, 65, 21 120, https://doi.org/10.3402/tellusb.v65i0.21120, 2013.

Kaimal, J. C., & Finnigan, J. J. (1994). Atmospheric boundary layer flows: their structure and measurement. Oxford university press.

Patton, E. G., Sullivan, P. P., Shaw, R. H., Finnigan, J. J., & Weil, J. C. (2016). Atmospheric stability influences on coupled boundary layer and canopy turbulence. Journal of the Atmospheric Sciences, 73(4), 1621-1647.